# BraVE: Offline Reinforcement Learning for Discrete Combinatorial Action Spaces

**Matthew Landers**[1][*], **Taylor W. Killian**[2], **Hugo Barnes**[1], **Thomas Hartvigsen**[1], **Afsaneh Doryab**[1]

[1]University of Virginia, [2]MBZUAI

## Abstract

Offline reinforcement learning in high-dimensional, discrete action spaces is challenging due to the exponential scaling of the joint action space with the number of sub-actions and the complexity of modeling sub-action dependencies. Existing methods either exhaustively evaluate the action space, making them computationally infeasible, or factorize Q-values, failing to represent joint sub-action effects. We propose **Bra**nch **V**alue **E**stimation (BraVE), a value-based method that uses tree-structured action traversal to evaluate a linear number of joint actions while preserving dependency structure. BraVE outperforms prior offline RL methods by up to $20\times$ in environments with over four million actions. [2]

## 1 Introduction

Offline reinforcement learning (RL) enables agents to learn decision-making policies from fixed datasets, avoiding the risks and costs inherent in online exploration [21, 22]. Existing methods have shown strong performance in low-dimensional discrete settings and continuous control tasks [1, 12, 13, 15, 19, 20]. However, many real-world decisions require selecting actions from high-dimensional, discrete spaces. In healthcare, for example, practitioners must choose among thousands of possible combinations of procedures, medications, and tests at each decision point [33].

Such settings give rise to *combinatorial* action spaces, where the number of actions grows exponentially with dimensionality, scaling as $\prod_{d=1}^{N} m_d$ for $N$ sub-action dimensions. In offline reinforcement learning, this structure induces two primary challenges: evaluating and optimizing over a large, discrete action space is computationally demanding; and making accurate decisions requires modeling sub-action dependencies from fixed datasets. Standard value-based methods can, in principle, capture sub-action dependencies but require evaluating or maximizing the Q-function over the full action space, which is intractable in high-dimensional settings. Conversely, recent approaches [3, 27] factorize the Q-function under a conditional independence assumption, reducing computational cost but limiting expressivity by precluding the modeling of sub-action interactions, which are critical in many real-world domains.

We introduce **Bra**nch **V**alue **E**stimation (BraVE), a value-based offline RL method for combinatorial action spaces that avoids the scalability–expressivity tradeoff of prior approaches. BraVE imposes a tree structure over the action space and uses a neural network to guide traversal, evaluating only a linear number of candidate actions while preserving sub-action dependencies. To support this structured selection process, the Q-function is trained with a behavior-regularized temporal difference (TD) loss and a branch value propagation mechanism.

Our experiments show that BraVE consistently outperforms state-of-the-art baselines across a suite of challenging offline RL tasks with combinatorial action spaces containing up to 4 million discrete

---

[*]mlanders@virginia.edu

[2]Code is available at `https://github.com/matthewlanders/BraVE`

39th Conference on Neural Information Processing Systems (NeurIPS 2025).

actions. In high-dimensional environments with strong sub-action dependencies, **BraVE improves average return by up to** $20\times$ **over state-of-the-art offline RL methods**. While baseline performance degrades with increasing sub-action dependencies or action space size, BraVE maintains stable performance.

Our contributions are as follows:

1. We propose a behavior-regularized TD loss that captures sub-action dependencies by evaluating complete actions rather than marginal components in combinatorial spaces.

2. We introduce **BraVE**, an offline RL method for discrete combinatorial action spaces that captures sub-action interactions and scales to high-dimensional settings via Q-guided traversal over a tree-structured action space.

3. We empirically show that BraVE outperforms state-of-the-art baselines in environments with combinatorial action spaces, maintaining high returns and stable learning as both action space size and sub-action dependencies increase.

## 2 Preliminaries

Reinforcement learning problems can be formalized as a Markov Decision Process (MDP), $\mathcal{M} = \langle \mathcal{S}, \mathcal{A}, p, r, \gamma, \mu \rangle$ where $\mathcal{S}$ is a set of states, $\mathcal{A}$ is a set of actions, $p : \mathcal{S} \times \mathcal{A} \times \mathcal{S} \rightarrow [0, 1]$ is a function that gives the probability of transitioning to state $s'$ when action $a$ is taken in state $s$, $r : \mathcal{S} \times \mathcal{A} \rightarrow \mathbb{R}$ is a reward function, $\gamma \in [0, 1]$ is a discount factor, and $\mu : \mathcal{S} \rightarrow [0, 1]$ is the distribution of initial states. A policy $\pi : \mathcal{S} \rightarrow \mathbb{P}(\mathcal{A})$ is a distribution over actions conditioned on a state $\pi(a \mid s) = \mathbb{P}[a_t = a \mid s_t = s]$.

While the standard MDP formulation abstracts away the structure of actions in $\mathcal{A}$, we explicitly assume that the action space is combinatorial; that is, $\mathcal{A}$ is defined as a Cartesian product of sub-action spaces. More formally, $\mathcal{A} = \mathcal{A}_1 \times \mathcal{A}_2 \times \cdots \times \mathcal{A}_N$, where each $\mathcal{A}_d$ is a discrete set. Consequently, $\mathbf{a}_t$ is an $N$-dimensional vector wherein each component is referred to as a sub-action.

The agent's goal is to learn a policy $\pi^*$ that maximizes cumulative discounted returns:

$$\pi^* = \arg \max_{\pi} \mathbb{E}_{\pi} \left[ \sum_{t=0}^{H} \gamma^t r(s_t, a_t) \right] \; ,$$

where $s_0 \sim \mu$, $a_t \sim \pi(\cdot \mid s_t)$, and $s_{t+1} \sim p(\cdot \mid s_t, a_t)$.

In offline RL, the agent learns from a static dataset of transitions $\mathcal{B} = \{(s_t, a_t, r_t, s_{t+1})^i\}_{i=0}^{Z}$ generated by, possibly, a mixture of policies collectively referred to as the behavior policy $\pi_\beta$.

Like many recent offline RL methods, our work uses approximate dynamic programming to minimize temporal difference error (TD error) starting from the following loss function:

$$L(\theta) = \mathbb{E}_{\mathcal{B}} \left[ \left( r + \gamma \max_{a'} Q(s', a'; \theta^-) - Q(s, a; \theta) \right)^2 \right] \; , \tag{1}$$

where the expectation is taken over transitions $(s, a, r, s')$ sampled from the replay buffer $\mathcal{B}$, $Q(s, a; \theta)$ is a parameterized Q-function that estimates the expected return when taking action $a$ in state $s$ and following the policy $\pi$ thereafter, and $Q(s, a; \theta^-)$ is a target network with parameters $\theta^-$, which is used to stabilize learning.

For out-of-distribution actions $a'$, Q-values can be inaccurate, often causing overestimation errors due to the maximization in Equation (1). To mitigate this effect, offline RL methods either assign lower values to these out-of-distribution actions via regularization or directly constrain the learned policy. For example, TD3+BC [13] adds a behavior cloning term to the standard TD3 [14] loss:

$$\pi = \arg \max_{\pi} \mathbb{E}_{(s,a) \sim \mathcal{B}} \left[ \lambda Q(s, \pi(s)) - (\pi(s) - a)^2 \right] \; , \tag{2}$$

where $\lambda$ is a scaling factor that controls the strength of the regularization.

More recently, implicit Q-learning (IQL) [19] used a SARSA-style TD backup and expectile loss to perform multi-step dynamic programming without evaluating out-of-sample actions:

$$L(\theta) = \mathbb{E}_{\mathcal{B}} \left[ \left( r + \gamma \max_{a' \in \Omega(s)} Q(s', a'; \theta^-) - Q(s, a; \theta) \right)^2 \right] \; , \tag{3}$$

where $\Omega(s) = \{a \in A \mid \pi_\beta(a \mid s) > 0\}$ are actions in the support of the data.

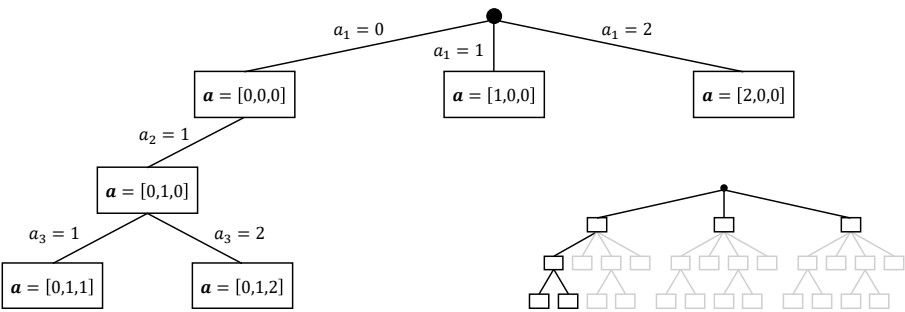

Figure 1: BraVE's tree representation for a 3-dimensional combinatorial action $\mathbf{a} = [a_1, a_2, a_3]$, where each sub-action $a_i \in \{0, 1, 2\}$. Each node encodes a complete action vector, with explicitly chosen sub-actions set according to the traversal path from the root, and all remaining dimensions filled with a default value (here, 0). At depth $k$, the value of sub-action $a_k$ is selected, with sibling nodes differing only in that dimension.

## 3 Branch Value Estimation (BraVE)

The Q-function, $Q(s, \mathbf{a})$ in Equation (1), encodes long-term value by mapping each state–action pair to its expected return. When actions comprise multiple sub-actions $\mathbf{a} = [a_1, \ldots, a_N]$, $Q(s, \mathbf{a})$ implicitly captures sub-action dependencies by accounting for interactions that influence both immediate rewards and future transitions, making it a natural target for learning in structured or combinatorial action spaces. However, in combinatorial action spaces, the exponential growth of joint actions renders both forward passes and Q-value estimation computationally intractable [30, 34].

Factorizing the Q-function across action dimensions [3, 27] is a common strategy for improving tractability. While this reduces computational cost, it sacrifices expressivity by ignoring sub-action dependencies, leading to estimation errors when sub-actions jointly influence transitions or rewards. In such settings, including real-world domains like treatment recommendation, summing marginal Q-values can produce divergent estimates (Appendix A).

We introduce **Bra**nch **V**alue **E**stimation (BraVE), an offline RL method that avoids this scalability–expressivity tradeoff. BraVE assigns values to complete action vectors $Q(s, \mathbf{a})$, preserving their interdependencies, but avoids exhaustively scoring every action by imposing a tree structure over the space and using a neural network to guide traversal that selectively evaluates a small subset of actions. This results in only a linear number of evaluations per decision, enabling BraVE to scale to high-dimensional combinatorial settings without sacrificing value function fidelity.

### 3.1 Tree Construction

BraVE imposes a tree structure over the combinatorial action space, assigning a complete $N$-dimensional action vector to each node. This vector is constructed by combining the sub-action values explicitly selected along the path from the root with predefined default values (e.g., $a_i = 0$ in the binary case) for all unassigned dimensions.

The tree originates from a root node corresponding to the start of traversal, where no sub-actions have been selected. From this root, branches extend to child nodes by explicitly assigning a value $a_k \in \mathcal{A}_k$ to the $k$-th sub-action dimension $d_k$. A node at depth $k$ therefore represents a complete action vector in which the first $k$ sub-actions $(a_1, \ldots, a_k)$ have been explicitly set, and the remaining sub-actions $(a_{k+1}, \ldots, a_N)$ retain their default values. Sibling nodes at depth $k$ differ from their parent only in the newly assigned component, and differ from each other only in their choice of $a_k$. That is, they differ only in the $k$-th component of the complete action vector they represent, as illustrated in Figure 1. This dimension-by-dimension construction ensures that reaching a leaf node, where all $N$ sub-action values have been explicitly assigned, requires traversing at most $N$ levels.

While this specific $N$-depth, dimension-ordered tree is adopted for its computational efficiency, the general BraVE framework places no constraints on the tree structure. Any tree in which each node represents a complete joint action, and in which root-to-leaf paths systematically cover the action space, is a valid instantiation. For example, a flat tree with a single level and $|\mathcal{A}|$ children — one for each possible joint action — recovers the naive alternative of exhaustively evaluating

$Q(s, \mathbf{a})$ over the entire space. Our implementation, however, deliberately uses the deeper, $N$-level structure to guarantee a number of node evaluations linear in $N$, which is crucial for tractability in high-dimensional spaces.

This structural flexibility extends beyond tree depth and layout. The ordering of sub-actions, the choice of default values, and the tree topology are all implementation details rather than core components of the method. Regardless of the selected configuration, however, the chosen structure should remain fixed throughout training to ensure consistent semantics for traversal and value estimation. The tree itself may be constructed dynamically at each timestep or precomputed and cached.

BraVE imposes no restrictions on the cardinality of individual sub-actions; each $a_d$ may be drawn from an arbitrary discrete set $\mathcal{A}_d$. For clarity, however, all examples and figures that follow assume a multi-binary action space, where each sub-action is either included ($a_i = 1$) or excluded ($a_i = 0$).

## 3.2 Tree Sparsification

To mitigate overestimation error and stabilize learning, BraVE introduces an inductive bias by **sparsifying the action space tree**, restricting it to include only actions observed in the dataset $\mathcal{B}$. This idea is conceptually related to BCQ [15], which constrains policy evaluation to actions deemed plausible by a learned generative model. BraVE, by constrast, enforces this constraint structurally by pruning branches from the tree that correspond to implausible or unsupported sub-action combinations.

This design is particularly effective in real-world settings where certain sub-actions are incompatible and rarely, if ever, appear together. For example, in treatment planning or medication recommendation, specific drug combinations may be avoided due to known adverse interactions. By limiting the tree to feasible actions observed in the data, BraVE avoids evaluating unrealistic behaviors, reduces computational overhead, and curbs value inflation driven by unsupported extrapolation.

## 3.3 Q-Guided Tree Traversal

To compute the greedy action $\hat{\mathbf{a}}' = \arg\max_{\mathbf{a}'} Q(s', \mathbf{a}'; \theta^-)$ required by Equation (1), BraVE traverses the action space tree using a neural network $f(\cdot, \cdot; \theta^-)$ to guide decisions at each node. At a node $\mathbf{a}_{\text{node}}$, the network receives as input the current state $s$, which may be discrete or continuous, and the $N$-dimensional action vector corresponding to $\mathbf{a}_{\text{node}}$. It outputs a scalar node Q-value $q = Q(s, \mathbf{a}_{\text{node}}; \theta^-)$ and a vector of *branch values* $\mathbf{v} \in \mathbb{R}^{M_{\text{children}}^{\max}}$. While $q$ reflects the value of selecting the current node's complete action, the branch values $\mathbf{v}$ estimate the maximum Q-value obtainable in the subtree rooted at the $j$-th child. The network produces both $q$ and $\mathbf{v}$ in a single forward pass, with the branch values $\mathbf{v}$ representing a learned estimate of each subtree's potential.

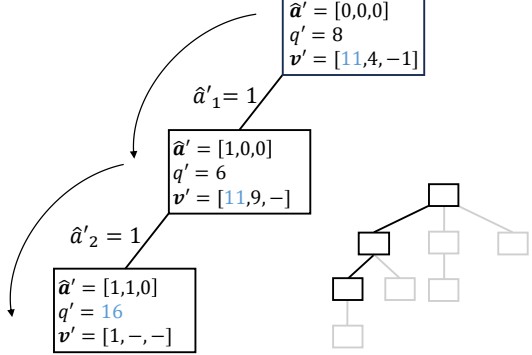

Figure 2: BraVE traversal in a 3-D binary action space (full tree shown bottom-right). Starting from the root $[0, 0, 0]$, the agent selects $\hat{a}'_1 = 1$ since its branch value (11) exceeds both those of alternative children $(4, -1)$ and the root's Q-value (8). Traversal proceeds until reaching $[1, 1, 0]$, where the Q-value (16) exceeds the child's branch value (1); a terminal condition. Masked values $(-)$ are ignored.

Importantly, the network $f$ is a standard MLP requiring no specialized architecture. The tree is not encoded into the network input; instead, it is an external structure that organizes the $\arg\max$ computation.

Though the number of children can vary across nodes, the network output dimensionality remains fixed. Specifically, $f$ returns one scalar for the current node and $M_{\text{children}}^{\max} = \max_d |\mathcal{A}_d|$ branch values. For example, if the maximum sub-action cardinality across all dimensions is three, the network produces four outputs total. Following standard practice in structured prediction and reinforcement learning [18, 26] masking is applied to ignore unused entries in $\mathbf{v}$ for nodes with fewer than $M_{\text{children}}^{\max}$ children.

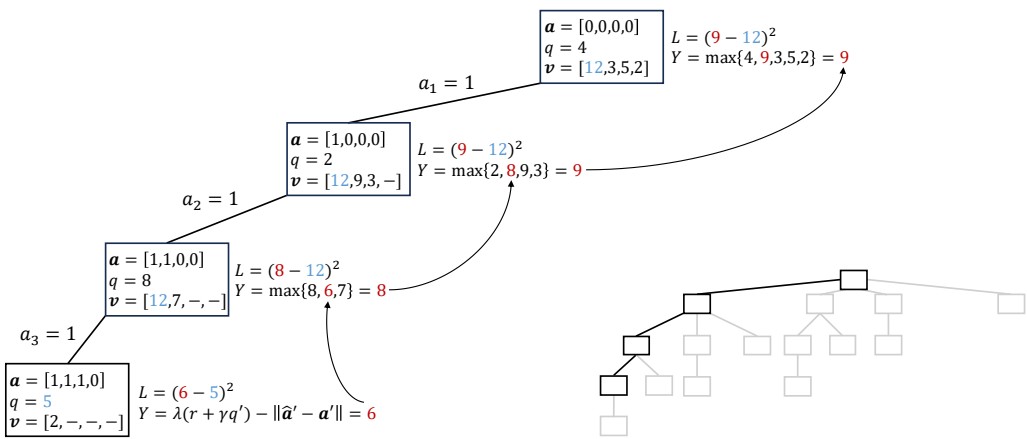

Figure 3: Example of loss propagation in a 4-D binary action space (full tree shown bottom-right). Starting from the node $[1, 1, 1, 0]$ (bottom left), the target (Equation 5) is propagated to its parent $[1, 1, 0, 0]$. The new target is computed as the maximum of the propagated value, the parent's own Q-value, and the branch values of alternative child nodes. This process recurses up the tree to compute all node losses.

Traversal begins at the root node. At each step, the network evaluates $(q, \mathbf{v})$ at the current node. If $q \geq \max_j v_j$, traversal terminates and the corresponding action $\mathbf{a}_{\text{node}}$ is returned as $\hat{\mathbf{a}}'$. Otherwise, the algorithm proceeds to the child indexed by $\arg\max_j v_j$. This greedy descent continues until either the termination condition is satisfied or a leaf node is reached. Notably, traversal may terminate at internal nodes rather than always proceeding to a fully specified leaf. A full example of this process is shown in Figure 2.

This traversal mechanism enables BraVE's computational efficiency: it evaluates a single node per sub-action dimension, resulting in $\mathcal{O}(N)$ complexity — a factor of $\prod_d |A_d|/N$ fewer evaluations than exhaustive scoring. The tree serves as the structural scaffold that makes this targeted, linear-time traversal possible.

### 3.4 Loss Computation

BraVE is trained using a behavior-regularized temporal difference (TD) loss that penalizes value estimates for actions unlikely under the dataset. Given a transition $(s, \mathbf{a}, r, s', \mathbf{a}')$ sampled from the replay buffer $\mathcal{B}$, the TD target is computed using the action $\hat{\mathbf{a}}' = \arg\max_{\mathbf{a}'} Q(s', \mathbf{a}'; \theta^-)$ selected via the tree traversal procedure described in Section 3.3. The loss is defined as:

$$L_{TD}(\theta) = \mathbb{E}_{(s,\mathbf{a},r,s',\mathbf{a}')\sim\mathcal{B}} \left[ \left( \lambda \left( r + \gamma Q(s', \hat{\mathbf{a}}'; \theta^-) \right) - \|\hat{\mathbf{a}}' - \mathbf{a}'\| - Q(s, \mathbf{a}; \theta) \right)^2 \right], \quad (4)$$

where $\lambda$ is a regularization coefficient and $\|\hat{\mathbf{a}}' - \mathbf{a}'\|$ penalizes deviation from the behavior action, following principles introduced in TD3+BC (Equation 2).

We combine this behavior-regularized TD loss with a branch value supervision loss $L_{\text{BraVE}}$, resulting in a total objective $L = \alpha L_{\text{TD}} + L_{\text{BraVE}}$, where $\alpha$ controls the relative weighting of the TD term. To compute $L_{\text{BraVE}}$, we begin from a node $\mathbf{a}$ sampled from $\mathcal{B}$ and reuse the TD target

$$Y = \lambda \left( r + \gamma Q(s', \hat{\mathbf{a}}'; \theta^-) \right) - \|\hat{\mathbf{a}}' - \mathbf{a}'\| \quad (5)$$

as the supervisory signal. This target is then propagated recursively up the tree. At each step, the target is used to update the parent node's estimate for the corresponding branch value. The propagated target is updated at each level to reflect the maximum of $Y$ and the existing sibling branch values, ensuring consistency with the max-based traversal logic. This process continues up to the root, training the branch values $\mathbf{v}$ at each node through direct supervision from $L_{\text{BraVE}}$ to estimate the highest Q-value accessible through its subtree. The full $L_{\text{BraVE}}$ computation procedure is provided in Algorithm 1 and illustrated in Figure 3.

The recursive mechanism underlying $L_{\text{BraVE}}$ stabilizes learning by propagating training signals not only to the node corresponding to the sampled action but also to every ancestor on its path. Consequently, internal nodes — including those not directly sampled — receive gradient updates. Because all branches share a single global Q-network, these updates generalize across similar states and actions, enabling BraVE to form reliable estimates even in rarely-sampled regions of the tree.

### 3.5 Enhancements for Stability and Policy Quality

We mitigate the sensitivity of BraVE's hierarchical structure to inaccurate branch value estimates using two complementary mechanisms: a **depth penalty** applied *during training* and **beam search** used *during inference*. During training, we apply a depth-based weighting factor $\delta$ in the BraVE loss computation. Because the branch value target is propagated upward from a sampled action node to the root, this weighting amplifies the influence of errors closer to the root, where mistakes propagate to more of the tree. At inference time, we use beam search to improve action selection robustness. Instead of committing to a single greedy path, the algorithm retains the top-$W$ actions (ranked by predicted values) at each tree level. The final action is selected from the union of all beams, allowing for broader exploration of high-value combinations and improving policy quality.

---

**Algorithm 1** Compute BraVE Loss

**Require:**
    $f(\theta)$: neural network with parameters $\theta$
    $f(\theta^-)$: target network with parameters $\theta^-$
    $\{s, \mathbf{a}, r, s', \mathbf{a}'\}$: transition from $\mathcal{B}$
    $\hat{\mathbf{a}}'$: action selected via tree traversal given $s$
1: $(q, \mathbf{v}) \leftarrow f(s, \mathbf{a}; \theta)$
2: $(q', \mathbf{v}') \leftarrow f(s', \hat{\mathbf{a}}'; \theta^-)$
3: $Y \leftarrow \lambda(r + \gamma q') - \|\hat{\mathbf{a}}' - \mathbf{a}'\|$
4: total loss $\leftarrow (q - Y)^2$
5: node $\leftarrow \mathbf{a}$
6: $d \leftarrow 1$
7: **while** node is not *null* **do**
8:     parent $\leftarrow$ GETPARENT(node)
9:     $q, \mathbf{v} \leftarrow f(s, \text{parent}; \theta)$
10:    children $\leftarrow$ GETCHILDREN(parent)
11:    $i \leftarrow$ index of node in children
12:    loss $\leftarrow ((\mathbf{v}[i] - Y) * \delta d)^2$
13:    total loss $\leftarrow$ total loss + loss
14:    $\mathbf{v}[i] \leftarrow Y$
15:    $Y \leftarrow \max(q, \mathbf{v})$
16:    node $\leftarrow$ parent
17:    $d \leftarrow d + 1$
18: **end while**
19: **return** total loss$/d$

---

## 4 Experimental Evaluation

We evaluate BraVE in the **Co**mbinatorial **N**avigation **E**nvironment (CoNE), a high-dimensional discrete control domain designed to stress-test policy learning under large action spaces and sub-action dependencies. In CoNE, actions are formed by selecting subsets of atomic motion primitives (sub-actions), corresponding to directional moves along orthogonal axes. The agent chooses which primitives to activate at each timestep, resulting in a combinatorial action space of size $|\mathcal{A}| = 2^{2D}$, where $D$ is the number of spatial dimensions.

The agent starts from a fixed origin $s_0$ and must reach a designated goal $g$. At each timestep, it receives a negative reward $r = -\rho(s, g)$ proportional to its Euclidean distance from the goal. Episodes terminate upon reaching the goal (with reward $+10$) or falling into a pit, a failure state that incurs a penalty of $r = -10 \cdot \rho(s_0, g)$. This penalty structure ensures that failure is strictly worse than any successful trajectory, even those that are long or indirect.

CoNE exhibits both combinatorial complexity and tightly coupled action dynamics. Some sub-action combinations are synergistic, such as activating orthogonal directions to move diagonally, opposing movements cancel out, and others lead to failure by directing the agent into hazardous regions of the state space. These structured dependencies mean that the effectiveness of a sub-action often depends critically on the presence or absence of others.

This stands in contrast to discretized variants of popular continuous control environments [28, 31], where sub-actions require coordination but can be learned independently [3]. In CoNE, by contrast, sub-actions that are individually beneficial can be harmful in combination, an effect common in real-world settings such as drug prescription, where treatments may interact antagonistically (see Appendix A).

**Dataset Construction** We generate offline datasets using a stochastic variant of $A^*$. At each step, the optimal action is selected with probability 0.1, and a random valid action is chosen otherwise. This procedure yields a diverse mixture of trajectories with varying returns, including both near-optimal and suboptimal behavior. The resulting datasets reflect realistic offline settings in which learning must proceed from heterogeneous, partially optimal demonstrations.

**Baselines** We compare BraVE to two representative baselines: (1) Factored Action Spaces (FAS) [3, 27], the state-of-the-art for offline RL in combinatorial action spaces; and (2) Implicit Q-Learning

| $|\mathcal{A}|$ | BraVE | FAS | IQL |
|---|---|---|---|
| 16 | 1.5 $\pm$ 0.0 | 1.5 $\pm$ 0.0 | $-0.4 \pm 1.5$ |
| 64 | -0.4 $\pm$ 0.0 | -0.4 $\pm$ 0.0 | $-6.1 \pm 3.2$ |
| 256 | -2.0 $\pm$ 0.1 | $-2.3 \pm 0.6$ | $-9.8 \pm 4.5$ |
| 1024 | -3.4 $\pm$ 0.1 | $-10.8 \pm 2.3$ | $-13.1 \pm 5.8$ |
| 4096 | -6.9 $\pm$ 1.6 | $-7.3 \pm 3.1$ | $-13.5 \pm 5.7$ |
| $\sim$16k | -6.1 $\pm$ 0.4 | $-25.6 \pm 33.3$ | $-15.4 \pm 6.0$ |
| $\sim$65k | -8.2 $\pm$ 2.2 | $-24.4 \pm 10.4$ | $-27.0 \pm 11.5$ |
| $\sim$260k | -13.8 $\pm$ 5.4 | $-42.2 \pm 32.3$ | $-48.4 \pm 17.7$ |
| $\sim$1M | -9.6 $\pm$ 1.2 | $-21.4 \pm 18.2$ | $-53.7 \pm 31.0$ |
| $\sim$4M | -18.6 $\pm$ 8.3 | $-33.9 \pm 27.0$ | $-66.9 \pm 31.6$ |

Table 1: In environments with non-factorizable reward structures and no sub-action dependencies BraVE and FAS perform similarly in low-dimensional settings, but as action dimensionality increases, BraVE maintains high returns and stable policies, while FAS's performance deteriorates. IQL performs worst in all settings.

(IQL) [19], a strong general-purpose algorithm included to assess the need for methods purpose-built for combinatorial structures. Following prior work [27], we implement FAS using a factored variant of BCQ [15]. All methods are trained for 20,000 gradient steps and evaluated every 100 steps.

### 4.1 Performance Across Reward Structures and Sub-Action Dependencies

FAS has shown strong performance in offline RL for combinatorial domains [3, 27], however, its effectiveness depends on the assumption that rewards decompose cleanly across sub-actions and that dependencies among sub-actions are weak. These assumptions are often violated in real-world settings, where the effect of one sub-action may depend critically on the value of another. To evaluate BraVE in precisely these challenging conditions, we construct a suite of CoNE environments that systematically vary along both axes of complexity.

We measure performance across 10 randomly generated environments in each of two settings: the first includes non-factorizable reward structures without sub-action dependencies, and the second includes both non-factorizable rewards and dependent sub-actions. For each task, we vary dimensionality from 2-D ($|\mathcal{A}| = 16$) to 11-D ($|\mathcal{A}| > 4$ million), where each additional dimension introduces two new sub-actions. Each environment has size 5 in every dimension. The agent starts in the top-left corner and must reach a goal in the bottom-right. Results are averaged over five random seeds.

**Non-Factorizable Reward Structures Only**   We begin with environments that contain no pits. In these settings, the transition dynamics are factorizable across sub-actions. However, the reward function cannot be linearly decomposed. As shown in Table 1, FAS performs comparably to BraVE in low-dimensional tasks but degrades as dimensionality increases. Because the reward depends on the state induced by the full action vector, the FAS approach of assigning rewards to individual sub-actions introduces a modeling bias that grows with action dimensionality, causing its learned Q-values to diverge from the true values. BraVE avoids this instability by evaluating complete joint actions directly, maintaining high performance across all action-space sizes. IQL performs significantly worse than BraVE across all action space sizes, suggesting that methods not explicitly designed for combinatorial actions struggle even when transitions are factorizable. Complete learning curves are provided in Appendix B.1.

**Non-Factorizable Rewards and Sub-Action Dependencies**   We next evaluate performance under increasing sub-action dependency induced by hazardous transitions. In an 8-D environment ($|\mathcal{A}| = 65,536$), we vary the number of pits from 5% to 100% of the 6,561 **interior** (non-boundary) states, ensuring that there is always a feasible path to the goal along the boundary of the state space (where pits are never placed). As shown in Table 2, BraVE maintains stable performance across all pit densities. FAS, by contrast, degrades sharply once pits occupy half of the interior, and IQL declines steadily as dependency strength increases. These results highlight BraVE's ability to handle tightly coupled decision-making, even when adverse interactions between sub-actions are common. These

| Pit % | BraVE | FAS | IQL |
|---|---|---|---|
| 0 | -8.2 $\pm$ 2.2 | $-24.4 \pm 10.4$ | $-27.0 \pm 11.5$ |
| 5 | -19.0 $\pm$ 2.3 | $-78.4 \pm 99.7$ | $-85.8 \pm 45.0$ |
| 10 | -16.9 $\pm$ 3.3 | $-140.4 \pm 177.1$ | $-88.1 \pm 50.1$ |
| 25 | -42.9 $\pm$ 43.3 | $-178.6 \pm 212.8$ | $-82.8 \pm 52.0$ |
| 50 | -54.8 $\pm$ 47.9 | $-902.6 \pm 274.1$ | $-106.4 \pm 42.8$ |
| 75 | -42.9 $\pm$ 34.3 | $-942.8 \pm 278.1$ | $-110.0 \pm 34.3$ |
| 100 | -41.6 $\pm$ 22.4 | $-1131.4 \pm 0.0$ | $-112.4 \pm 22.8$ |

Table 2: BraVE maintains stable performance across environments with non-factorizable rewards and sub-action dependencies, whereas FAS exhibits a sharp decline when pits occupy at least half of the interior states. IQL performance also degrades as sub-action dependencies increase.

results are further contextualized by the normalized episode-length score reported in Appendix B.2, where complete learning curves are also provided.

**Sparse but Critical Sub-Action Dependencies**    Having established that BraVE performs well under strong sub-action coupling, we examine settings with only five pits, representing a negligible fraction of the state space in high dimensions. We continue scaling the state space until BraVE and FAS converge to similar average returns, allowing us to assess how each method handles **sparse but critical dependencies**. As shown in Table 3, BraVE performs reliably even when the state space is small and each action carries substantial risk. FAS, by contrast, requires a much larger state space before the influence of the pits becomes diluted and its performance approaches that of BraVE. This demonstrates that even a small number of pits is sufficient to invalidate naive factorizations. Complete learning curves are provided in Appendix B.3.

| $|\mathcal{S}|$ | | BraVE | FAS | IQL |
|---|---|---|---|---|
| 25 | | -7.5 $\pm$ 2.4 | $-531.5 \pm 31.4$ | $-12.1 \pm 5.8$ |
| 125 | | -2.9 $\pm$ 0.2 | $-579.8 \pm 7.2$ | $-14.1 \pm 13.6$ |
| 625 | | -5.6 $\pm$ 1.5 | $-480.3 \pm 152.9$ | $-22.4 \pm 20.6$ |
| 3125 | | -6.3 $\pm$ 0.7 | $-147.4 \pm 292.6$ | $-28.0 \pm 22.8$ |
| 15625 | | -12.4 $\pm$ 7.9 | $-18.9 \pm 4.6$ | $-37.8 \pm 17.4$ |

Table 3: When sub-action dependencies are sparse but critical, BraVE performs reliably even in small state spaces with high-risk actions. FAS approaches BraVE's performance only once the joint action space exceeds 15 thousand. IQL performs worse than BraVE in all settings.

## 4.2   Robustness to Action Order

BraVE's tree structure is defined by a fixed sequence of sub-action dimensions. To assess sensitivity to action ordering, we trained BraVE, FAS, and IQL using five random permutations of this sequence in the 8-D, 50% pit CoNE setting, which is empirically the most challenging 8-D variant (see Table 2). Results averaged over these permutations are reported in Table 4.

These findings are consistent with those in Table 2, showing that BraVE outperforms FAS by an order of magnitude and is roughly twice as effective as IQL. This demonstrates that BraVE's performance is robust to the ordering of sub-action dimensions and does not rely on a favorable structural arrangement.

## 4.3   Online Fine-Tuning

BraVE's data-driven tree sparsification (Section 3.2) inherently constrains action selection to observed behaviors, a principle shared by offline methods that model the behavior policy using generative approaches [15]. However, BraVE imposes this constraint structurally, eliminating the reliance on an explicit behavior model that can complicate online fine-tuning [23]. Consequently, BraVE can

| BraVE | FAS | IQL |
|---|---|---|
| -53.5 $\pm$ 29.1 | $-1103.6 \pm 51.4$ | $-96.2 \pm 51.8$ |

Table 4: Performance on the 8-D, 50% pit CoNE task, averaged over five random permutations of the sub-action dimensions. BraVE's effectiveness is not dependent on a specific tree construction.

be directly fine-tuned online without modification. Appendix C shows that BraVE matches IQL's performance during fine-tuning when both methods begin from comparable offline initialization.

### 4.4 Ablations and Hyperparameters

We conduct ablation and hyperparameter studies to quantify the contribution of each of BraVE's design choices. First, we examine the depth penalty $\delta$, which scales the loss by a node's position in the tree. A small penalty ($\delta = 1$) consistently yields the best results, confirming the importance of prioritizing accuracy near the root. Second, we vary the loss weight $\alpha$ to analyze the trade-off between the behavior-regularized TD loss and the BraVE loss. Performance remains stable across a range of $\alpha$ values, but omitting the TD term entirely ($\alpha = 0$) leads to significant degradation, particularly in higher-dimensional tasks. Third, we isolate the contribution of BraVE's tree structure by training a standard DQN baseline, which omits the hierarchical traversal and branch value propagation. This variant fails to learn a viable policy, confirming that BraVE's structured decomposition is critical to its success. Fourth, we evaluate the effect of beam width on performance and inference time. A width of 10 is sufficient for strong performance, with larger widths offering negligible gains at minimal runtime cost, reflecting the approach's practical efficiency. Full results are presented in Appendix D.

## 5 Related Work

### 5.1 Tree-based RL

Tree-based methods in RL have historically been applied to ordered decision processes. Most notably, Monte Carlo Tree Search (MCTS) [6], as used in AlphaZero [26], recursively selects actions using the PUCT algorithm [2]. However, its sequential structure makes it ill-suited for unordered action spaces like those considered here, where sub-actions must be selected simultaneously.

TreeQN [10] blends model-free RL with online planning by constructing a tree over learned state representations, refining value estimates via tree backups. Differentiable decision trees (DDTs) [25] enable gradient-based learning in RL by replacing the hard splits typical of decision trees with smooth transitions, which can later be discretized for interpretability. In the offline setting, Ernst et al. [9] apply tree-based regression (e.g., CART, Kd-trees) to approximate Q-functions via fitted Q-iteration.

### 5.2 Combinatorial Action Spaces

Many RL approaches have been developed for combinatorial action spaces, particularly in domain-specific contexts such as text-based games [16, 17, 34], routing [7, 24], TSP [4], and resource allocation [5]. These methods often depend on domain structure, whereas BraVE is domain-agnostic.

General-purpose approaches include distributed action representations [29], curriculum-based action space growth [11], and search-based amortized Q-learning [32], which replaces exact action maximization with optimization over sampled proposals. Wol-DDPG [8] embeds discrete actions into continuous spaces, though this approach has been found to be ineffective in unordered settings [5]. Notably, these are online approaches, and their application to offline settings is often non-trivial. For example, Zhao et al. [35] rely on state-action visitation counts, which are generally infeasible to obtain offline.

Closer to our setting, Tang et al. [27] and Beeson et al. [3] evaluate Factored Action Spaces (FAS in our experiments), which linearly decompose the Q-function by conditioning each term on a single sub-action. While this reduces dimensional complexity, it relies on sub-action independence; a condition that often fails in real-world domains. Crucially, these limitations are algorithm-agnostic and apply to factorized variants of all offline RL methods. BraVE, by contrast, structures the action space to retain sub-action dependencies without incurring exponential cost.

# 6 Discussion and Conclusion

In many real-world decision-making problems, combinatorial action spaces arise from the simultaneous selection of multiple sub-actions. Offline RL in these settings remains challenging due to the cost of optimization and the difficulty of modeling sub-action dependencies from fixed datasets. Standard methods do not scale, while factorized approaches improve tractability at the expense of expressivity. We present **Bra**nch **V**alue **E**stimation (BraVE), an offline RL method for discrete combinatorial action spaces. By structuring the action space as a tree, BraVE captures sub-action dependencies while reducing the number of actions evaluated per timestep, enabling scalability to large spaces. BraVE outperforms state-of-the-art baselines across environments with varying action space sizes and sub-action dependencies.

While BraVE demonstrates strong performance, several limitations and opportunities for future work remain. First, the tree structure is defined by a fixed ordering of the sub-action dimensions. While Section 4.2 demonstrates robustness to this choice in CoNE, future work could explore methods for learning or adaptively selecting the dimension sequence. Second, while BraVE's inference complexity is linear in the number of dimensions ($N$), the size of the network's output layer and the tree's branching factor grow with the sub-action cardinality ($|\mathcal{A}_d|$), which may impact performance on tasks with extremely high-cardinality discrete actions. Third, the recursive update for $L_{\text{BraVE}}$ requires backpropagation through all parent nodes of a sampled action, incurring a greater computational cost during training than a standard Q-learning update. Fourth, our empirical validation is confined to the synthetic CoNE benchmark; future work should assess BraVE's generalizability across a broader range of combinatorial tasks and real-world applications. Finally, adapting the BraVE framework to online learning is an important direction for future work, requiring the development of new mechanisms to manage the exploration-exploitation trade-off within the structured action space.

Despite these limitations, BraVE provides a flexible and extensible foundation for future research on combinatorial action spaces. We believe its applicability extends well beyond the specific setting explored in this paper. In particular, constrained RL presents a promising direction, as BraVE's tree structure naturally supports the exclusion of inadmissible action combinations through pruning.

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

# A Limitations of Standard Value-Based Estimation and Q-Function Factorization in Combinatorial Action Spaces with Sub-Action Dependencies

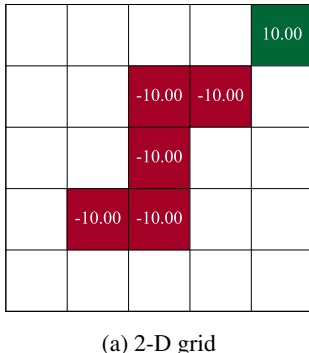

(a) 2-D grid

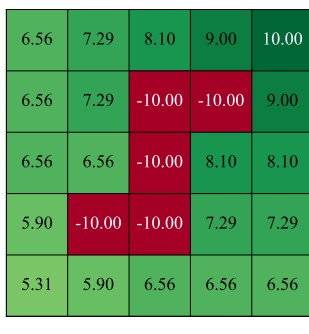

(b) Q-values

Figure 4: A 2-D grid with five pits and the true maximum Q-values in each state.

Consider a two-dimensional grid with five pits, which provides a simplified instance of CoNE, allowing us to clearly illustrate the limitations of standard methods. At each state, the agent chooses from 16 possible actions ($|\mathcal{A}| = 16$), corresponding to combinations of movement directions (e.g., $\emptyset$, [U], [UD], [UDL], [ULR], [UDLR], [D], [DR], ... ). Transitions yield a reward of $r = 0$ unless they lead to the goal ($r = 10$) or a pit ($r = -10$), as shown in Figure 4a.

Sub-action dependencies are inherently captured by the standard Q-function, as they affect both immediate rewards and future returns:

$$Q_\pi(s, a) = r(s, a) + \gamma \sum_{s' \in \mathcal{S}} p(s, a, s') \left( \sum_{a' \in \mathcal{A}} \pi(a' \mid s') Q_\pi(s', a') \right) .$$

Thus, the agent can learn the true maximum Q-value in each state (Figure 4b) and, consequently, recover the optimal policy.

Accurately estimating the Q-function is challenging in combinatorial action spaces, where the number of possible actions grows exponentially with the action dimension. This complexity limits the applicability of value-based RL methods, such as IQL, which require estimating values for all actions.

More specifically, IQL's policy loss is defined as:

$$L_\pi(\phi) = \mathbb{E}_{(s,a) \sim \mathcal{B}} \left[ \exp \left( \beta \left( Q(s, a; \theta^-) - V(s; \psi) \right) \right) \log \pi_\phi(a \mid s) \right] .$$

In discrete action settings, the policy $\pi_\phi(a \mid s)$ is typically parameterized as a flat categorical distribution. That is, a policy network maps the state $s$ to a vector of logits $\ell_\phi(s) \in \mathbb{R}^{|\mathcal{A}|}$, where $|\mathcal{A}|$ is the number of discrete actions. The softmax function then transforms these logits into a probability distribution:

$$\pi_\phi(a \mid s) = \frac{\exp(\ell_\phi(s)_a)}{\sum_{b=1}^{|\mathcal{A}|} \exp(\ell_\phi(s)_b)} .$$

This formulation requires a forward pass through the policy network to produce logits for all possible actions, incurring computational cost and modeling complexity that scale with $|\mathcal{A}|$.

BraVE evaluates only a subset of possible actions at each timestep, allowing it to exploit the Q-function's capacity to capture sub-action dependencies without predicting an exponential number of action values or logits simultaneously. Factored approaches (FAS) [3, 27], by contrast, simplify computation by linearly decomposing the Q-function, conditioning each component on a single sub-action and the full state. While this reduces the dimensional complexity of the action space, it imposes structural constraints on the reward and Q-function definitions:

$$r(s, a) = \sum_{d=1}^{D} r_d(s, a_d) \quad \text{and} \quad Q(s, a) = \sum_{d=1}^{D} q_d(s, a_d) , \tag{6}$$

that may limit expressiveness in settings with sub-action dependencies and cause the Q-function to *diverge*.

More specifically, when sub-actions are combined (e.g., "up" + "left"), FAS retrieves partial Q-values $q_1(s, \text{up})$ and $q_2(s, \text{left})$ from different next states, rather than from a single consistent next state determined by the combined action. This results in artificially inflated values, as the summations in Equation (6) incorrectly assume independence. Iterative Bellman updates, as used in Q-learning, amplify this inconsistency, ultimately causing Q-value divergence.

The independence assumption holds in discretized variants of popular continuous control environments, where sub-actions typically influence distinct parts of the state space [28, 31]. However, this assumption is violated in many real-world settings, a limitation explicitly acknowledged in the FAS literature [3, 27].

# B CoNE Learning Curves

We compare BraVE's performance to two state-of-the-art baselines: Factored Action Spaces (FAS) [27, 3], which learns linearly decomposable Q-functions for offline combinatorial action spaces, and Implicit Q-Learning (IQL) [19], a general-purpose offline RL method. All methods are evaluated across 20 instances of CoNE. In CoNE, the sizes of both the action and state spaces increase exponentially with the environment's dimensionality: a $D$-dimensional setting with $M$ positions per axis yields $|\mathcal{A}| = 2^{2D}$ joint actions and $|\mathcal{S}| = M^D$ distinct states. All experiments were conducted on a single NVIDIA A40 GPU using Python 3.9 and PyTorch 2.6.

In all curves, results are averaged over 5 seeds; shaded regions indicate one standard deviation.

## B.1 Non-Factorizable Reward Structures Only

In environments without pits, transition dynamics are factorizable across sub-actions. The reward function, however, remains non-linear and cannot be decomposed. Below we present the training curves corresponding to the results in Table 1.

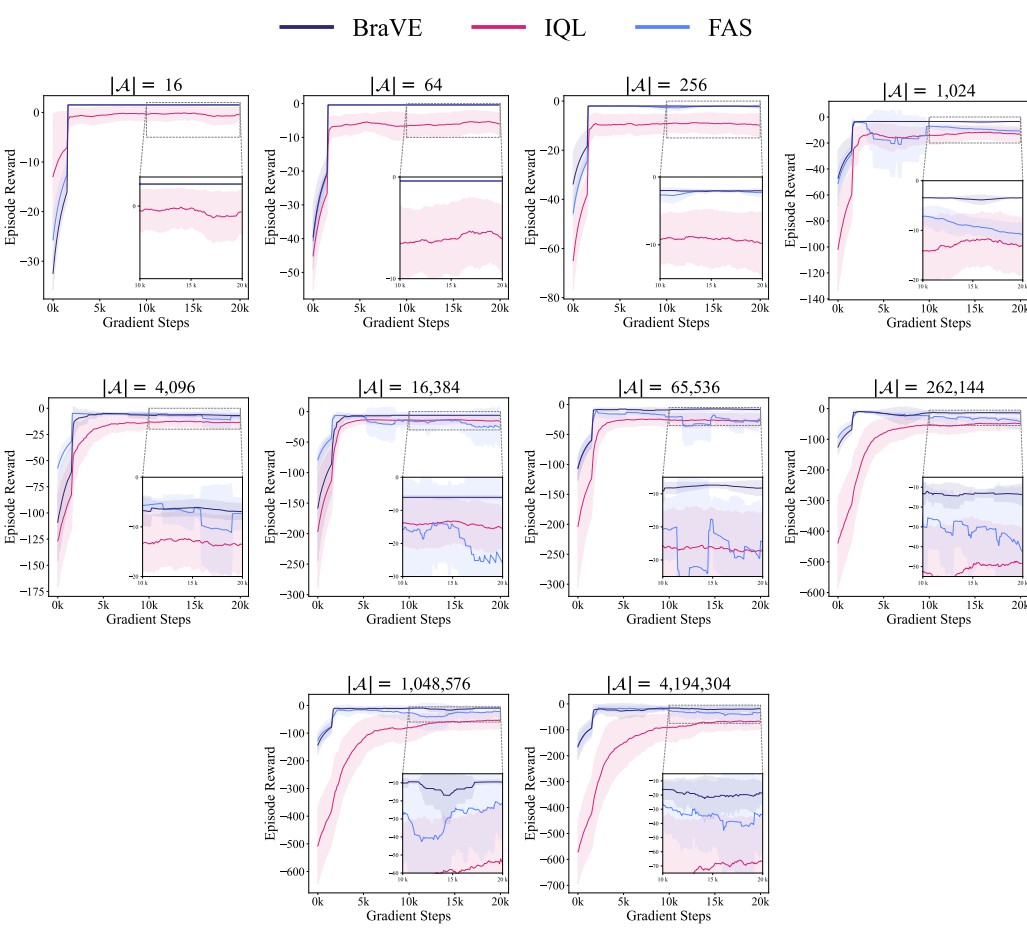

Figure 5: Learning curves for BraVE, FAS, and IQL in environments with non-factorizable reward structures but no sub-action dependencies.

In low-dimensional settings, FAS performs similarly to BraVE, but its performance degrades as dimensionality increases due to the bias introduced by linear Q-function decomposition. BraVE, by contrast, remains stable across all action space sizes by evaluating joint actions directly. IQL performs relatively poorly in all configurations, indicating that general-purpose offline RL methods struggle even when transition dynamics are factorizable.

## B.2 Non-Factorizable Reward Structures and Sub-Action Dependencies

To assess performance under increasing sub-action dependencies, we introduce hazardous transitions by varying the density of pits in an 8-D environment ($|\mathcal{A}| = 65,536$). The number of pits ranges from 5% to 100% of the 6,561 interior (non-boundary) states. Below we present the training curves and normalized episode-length scores corresponding to the results in Table 2.

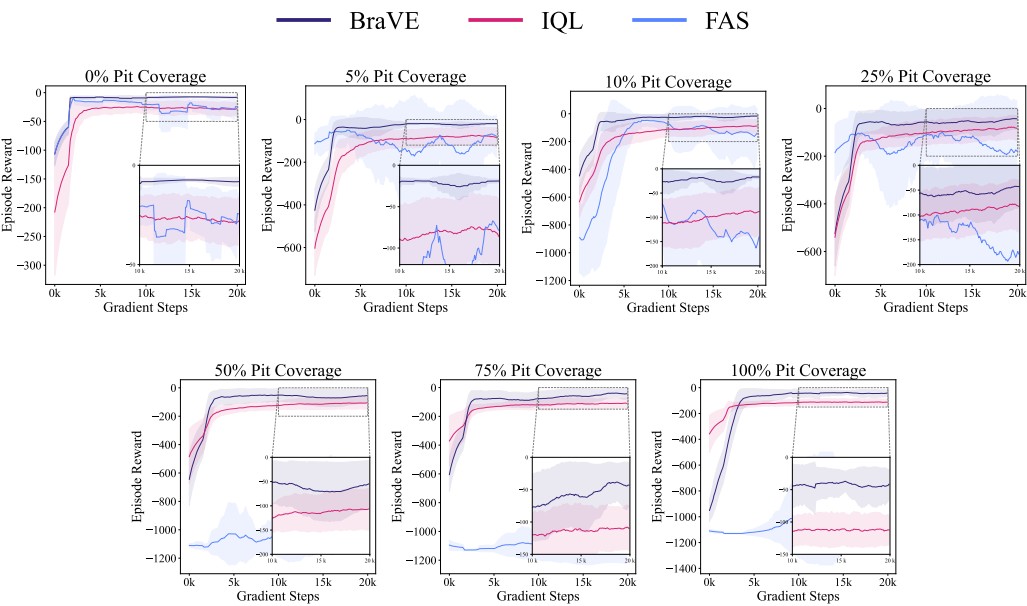

Figure 6: Learning curves for BraVE, FAS, and IQL in environments with non-factorizable reward structures and sub-action dependencies.

**Learning Curves** BraVE maintains stable returns across all pit densities, whereas FAS degrades sharply when pits cover at least half the interior states. IQL performance degrades as dependency strength increases. These results demonstrate BraVE's effectiveness in environments with tightly coupled sub-action effects and frequent adverse interactions.

**Normalized Episode-Length Score** To further contextualize performance, we compute a normalized episode-length score that places each method on a 0–100 scale, where 0 corresponds to a random policy and 100 to an oracle planner. It is defined as:

$$\text{score}_{\text{norm}} = 100 \cdot \frac{\text{length}_{\text{random}} - \text{length}_{\text{agent}}}{\text{length}_{\text{random}} - \text{length}_{\text{oracle}}} \; ,$$

where $\text{length}_{\text{agent}}$ is the average episode length of the evaluated policy, $\text{length}_{\text{random}}$ is the episode length of a random policy (which consistently times out at 100 steps), and $\text{length}_{\text{oracle}}$ is the optimal path length computed via A$^*$ search.

The normalized scores in Table 5 reinforce the findings from Table 2. BraVE consistently achieves scores above 70, indicating near-optimal behavior even in the most hazardous settings. FAS, by contrast, collapses to 0 — equivalent to a random policy — once dependencies become sufficiently strong.

| Pit % | BraVE | FAS | IQL |
|---|---|---|---|
| 0 | 99.3 | 89.6 | 88.0 |
| 5 | 92.8 | 57.3 | 52.9 |
| 10 | 94.1 | 20.2 | 51.5 |
| 25 | 78.5 | 0.0 | 54.7 |
| 50 | 71.4 | 0.0 | 40.6 |
| 75 | 78.5 | 0.0 | 38.4 |
| 100 | 79.3 | 0.0 | 37.0 |

Table 5: Normalized episode-length scores (0=random, 100=oracle) in the 8-D CoNE environment with varying pit densities. Scores show BraVE maintaining high performance while FAS's performance collapses to that of a random policy.

### B.3 Sparse but Critical Sub-Action Dependencies

To isolate the effect of sparse but critical sub-action dependencies, we consider environments with only five pits, representing a negligible fraction of the state space in higher dimensions. Starting in the 2-D environment, we progressively scale the state space until BraVE and FAS converge to similar average returns, allowing us to evaluate how each method handles **low-frequency but high-impact hazards**. Below we present the training curves corresponding to the results in Table 3.

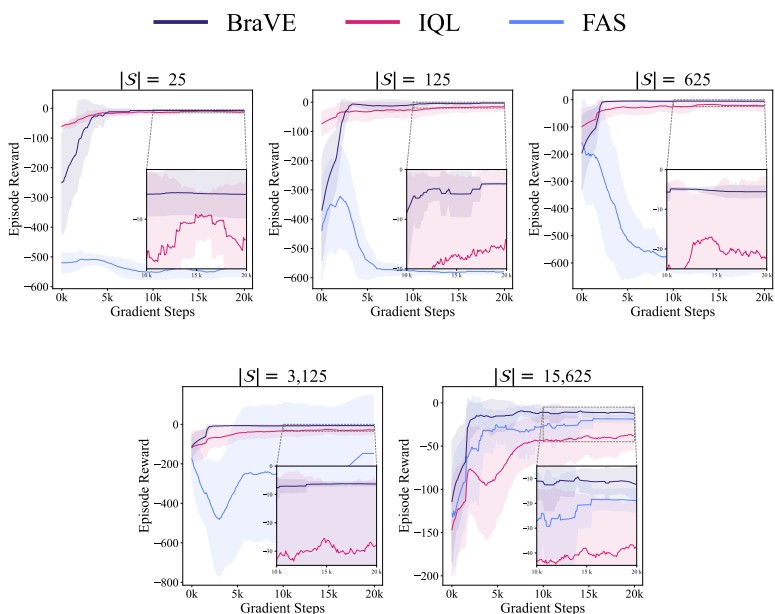

Figure 7: Learning curves for BraVE, FAS, and IQL in environments with sparse but critical sub-action dependencies.

BraVE performs reliably across all settings with sparse but critical sub-action dependencies even when the environment is small and each action carries significant risk. FAS, by contrast, requires much larger state spaces before the influence of the pits diminishes and its performance approaches that of BraVE. These findings indicate that even a small number of hazardous transitions can invalidate simplistic factorizations. IQL performs worse than BraVE in all settings.

# C Online Fine-Tuning After Offline Learning

To evaluate BraVE's capacity for online fine-tuning, we compare it to IQL, a method known for strong online adaptation following offline pretraining. FAS is excluded from this experiment because, as a factorized version of BCQ, it is not suited to online fine-tuning [23].

We conduct this experiment in an 8-D environment with 3,281 pits (50% of interior states), which is empirically the most challenging 8-D CoNE configuration for BraVE. Success in this setting best indicates generalizability to environments where BraVE demonstrates stronger performance.

Both methods are first trained offline for 1,500 gradient steps, after which BraVE achieves an average return of -226.1 compared to IQL's -259.4. We terminate offline training at this point, as BraVE consistently surpasses IQL beyond this threshold (see Figure 6). Online fine-tuning then proceeds for an additional 5,000 environment steps.

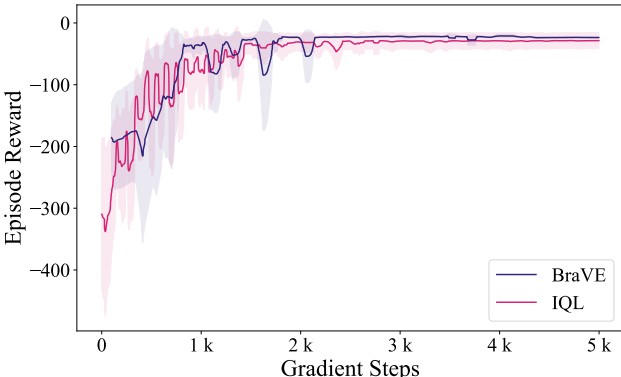

Figure 8: Learning curves for finetuning BraVE and IQL after 1,500 offline gradient steps.

As shown in Figure 8, the methods achieve comparable final performance during fine-tuning, with BraVE slightly outperforming IQL.

# D    Ablation and Hyperparameter Learning Curves

This section presents learning curves from four ablation and hyperparameter studies: (1) the effect of varying the depth penalty $\delta$, (2) BraVE's sensitivity to $\alpha$, the weight of the TD error in the total loss, (3) the importance of BraVE's tree structure, and (4) the effect of beam width on both return and computational overhead. A summary of these results appears in Figure 9, with full details provided in Appendices D.1, D.2, D.3, and D.4, respectively. All experiments are conducted in five-pit environments.

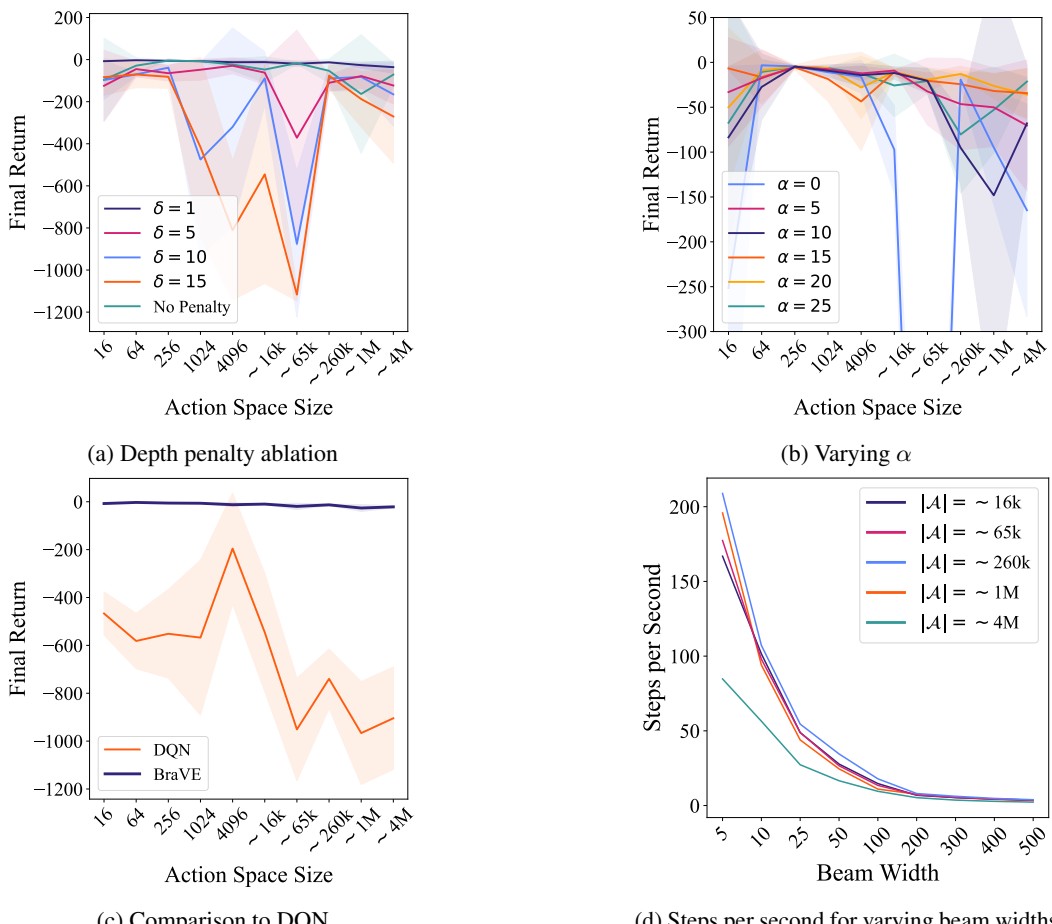

(a) Depth penalty ablation

(b) Varying $\alpha$

(c) Comparison to DQN

(d) Steps per second for varying beam widths

Figure 9: A small depth penalty ($\delta = 1$) yields the best performance across all environments (Figure 9a). Performance remains relatively stable across a range of $\alpha$ values; however, omitting the TD loss entirely ($\alpha = 0$) can result in suboptimal policies (Figure 9b). Although constraining the DQN to select actions within $\mathcal{B}$ introduces an inductive bias, it fails to produce a viable policy (Figure 9c). Beam width has minimal practical impact on runtime as inference remains under one second even with beam widths well above the optimal setting (Figure 9d).

## D.1 Depth Penalty Ablation

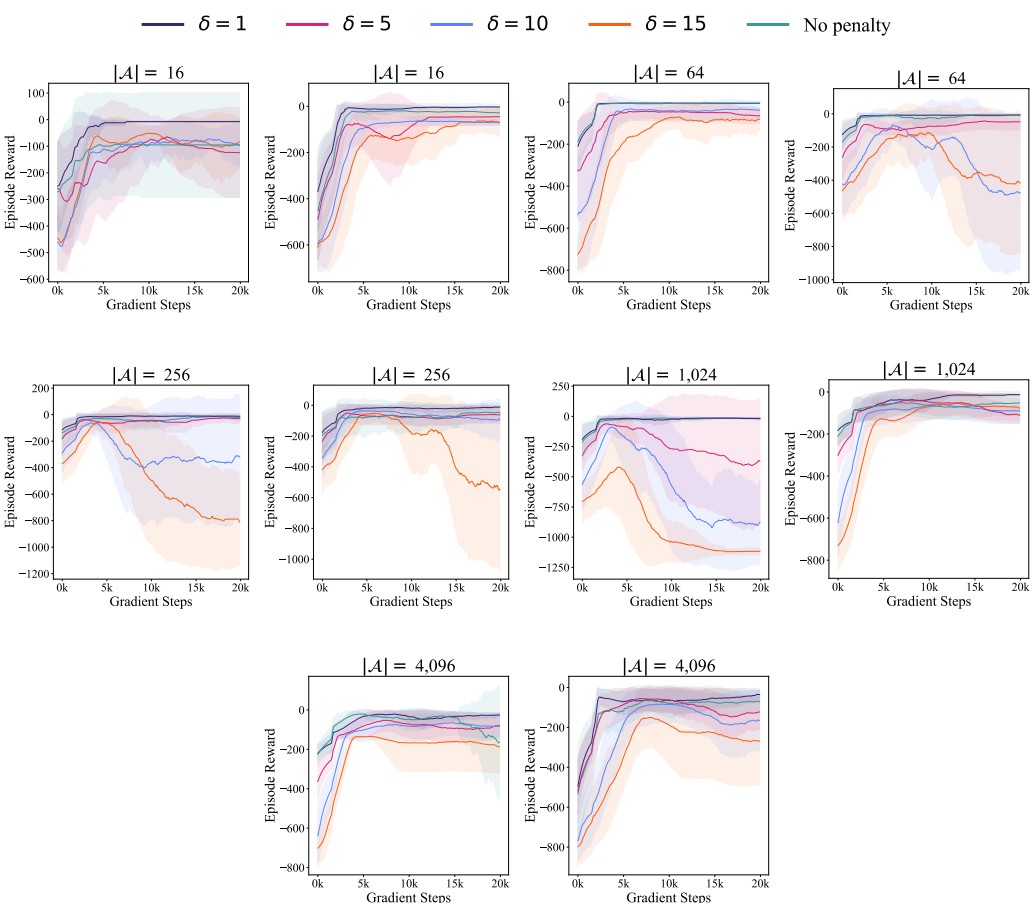

Figure 10: Learning curves for BraVE with varying depth penalty values ($\delta$).

While the depth penalty improves performance, it should remain small; $\delta = 1$ yields optimal results across nearly all environments.

## D.2 Varying TD Weight in Loss

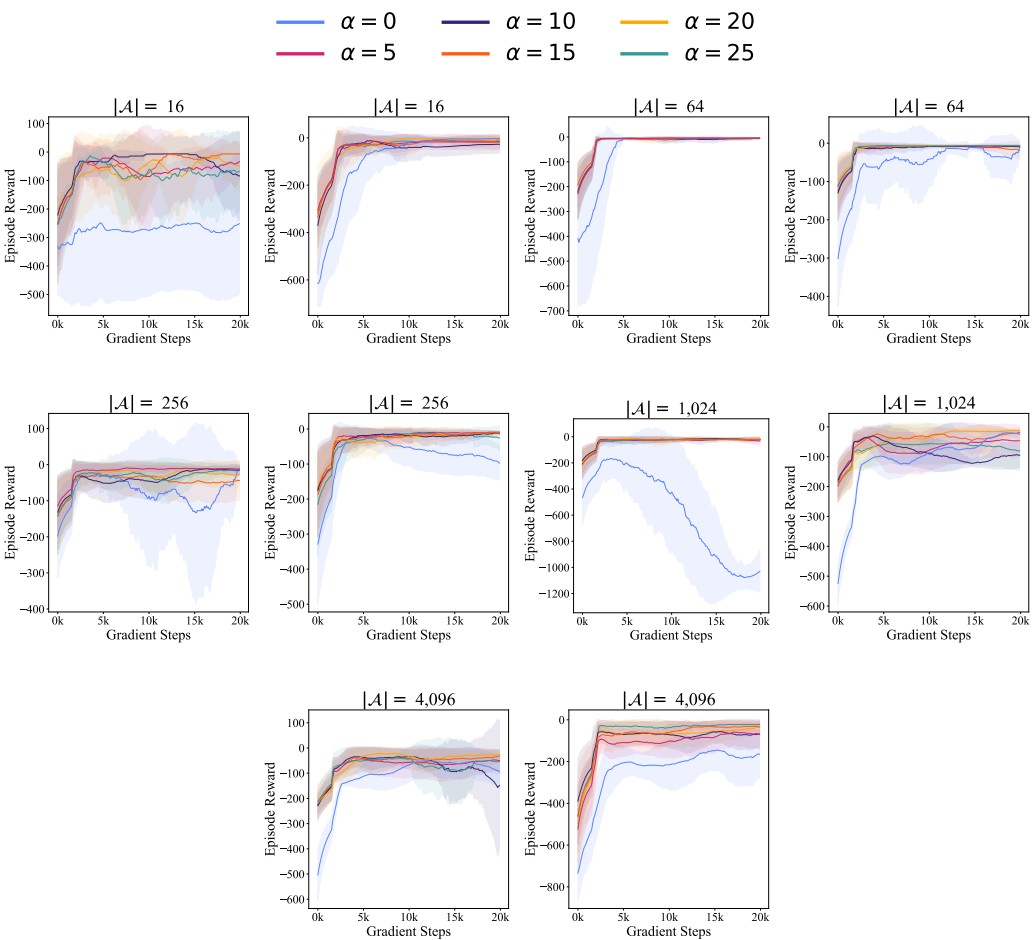

Figure 11: Learning curves for BraVE with varying TD loss weight values ($\alpha$).

Performance is relatively stable across $\alpha$ values in low-dimensional environments, with the exception of the 2-D case, where the high pit density makes accurate decision-making critical. In higher-dimensional settings, BraVE becomes more sensitive to the choice of $\alpha$. In particular, omitting the TD loss term ($\alpha = 0$) can result in catastrophic failures, especially in complex environments such as the 8-D setting.

### D.3 Comparison to DQN

To isolate the effect of BraVE's tree structure, we compare it to a DQN constrained to selecting actions from the dataset $\mathcal{B}$ and trained using BraVE's behavior-regularized TD loss (Equation 4).

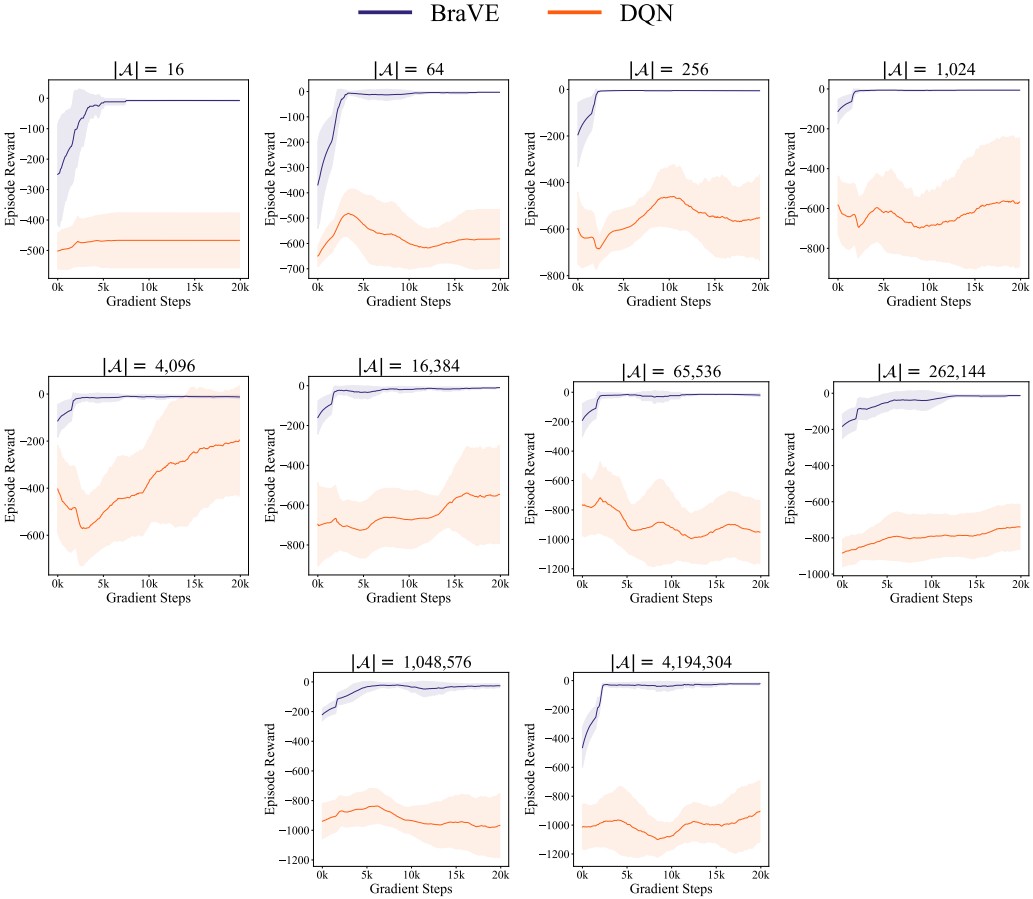

Figure 12: Learning curves for BraVE and a DQN constrained to select actions from $\mathcal{B}$ and trained with BraVE's behavior-regularized TD loss (Equation 4) — effectively, BraVE without tree-based action selection or branch value propagation.

Although the DQN is trained with the same behavior-regularized TD loss as BraVE and is restricted to selecting actions in $\mathcal{B}$, it fails to learn an effective policy. This suggests that the DQN struggles to capture sub-action dependencies, particularly in large action spaces. For example, in the 11-D environment, it must evaluate all 8,927 unique actions in $\mathcal{B}$ simultaneously. BraVE mitigates this complexity by organizing the action space as a tree, reducing the number of required predictions to a small subset of Q-values at each timestep.

## D.4   Beam Width

Because beam search is used only during policy extraction — after value estimates have been learned — it remains independent of the learning process. This decoupling allows for post-training optimization of beam width, provided the environment permits hyperparameter tuning. To evaluate the impact of beam width on both return and computational overhead, we measure return, environment steps per second, and total episode runtime for beam widths ranging from 5 to 500.

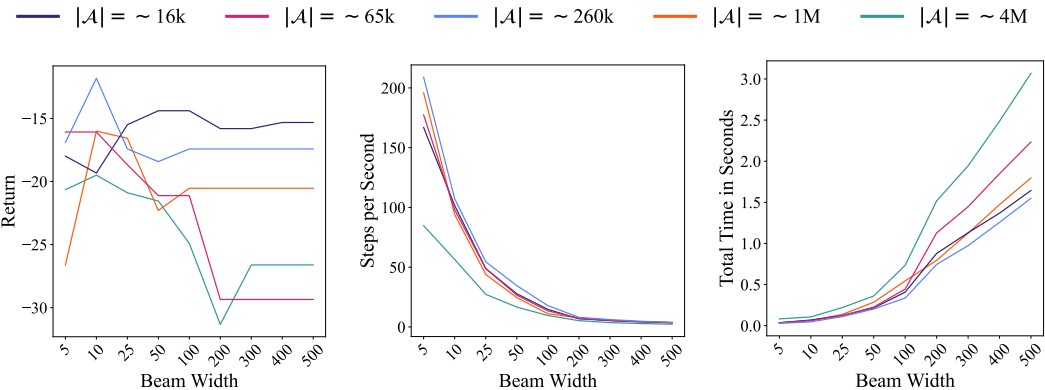

Figure 13: Effect of beam width on return and computational cost, measured in environment steps per second and total episode runtime.

A beam width of 10 is generally sufficient for near-optimal performance. Larger beams can degrade performance by exploring low-confidence regions of the tree, where Q-value estimates are less reliable. Narrower beams act as a form of regularization, limiting search to well-learned regions of the action space.

Beam width has negligible practical impact on runtime: inference times remain well below one second, even for beam widths far exceeding the optimal setting. These results are based on tests conducted on a 10-core CPU with 32 GB of unified memory and no GPU.

