# OpenReview forum: "BraVE: Offline Reinforcement Learning for Discrete Combinatorial Action Spaces"
_NeurIPS.cc/2025/Conference — NeurIPS 2025 poster_

### Official Review · Reviewer_BzAb · 2025-06-21

**Clarity:** 3
**Significance:** 2
**Originality:** 3
**Rating:** 4
**Confidence:** 3

**Summary:**

This paper proposed to handle combinatorial action spaces which can be very large in offline RL. The authors proposed to construct a tree to sequentially handle each action dimension relying on node Q values and branch V values. The authors further restricted the tree to only dataset actions to sparsify the tree to lower computational cost.

**Questions:**

Besides the questions in the weakness section, I am curious that can this sequential traversal approach be applied to softmax action selection?

**Ethical Concerns:**

["NO or VERY MINOR ethics concerns only"]

**Final Justification:**

The paper studies combinatorial action spaces which is of vital importance to RL. Despite some issues on clarification limitations such as error influence conditioned on tree level, I believe the paper offers contributions that will be of interest to the RL community. The authors engaged in discussion actively, and I have requested the authors to clearly point out the remaining issues since they do not overweigh the contributions.

**Limitations:**

No societal impact applicable. For limitations, the authors have discussed some limitations of the paper, as the tree traversal indeed relies on a fixed predefined ordering of sub-action dimensions.

**Quality:**

3

**Strengths And Weaknesses:**

**Strengths:**\

Handling combinatorial action spaces is of fundamental interest to RL. I think the paper is overall well-written and the tree-based method is principled. Positioning the tree structure in offline RL where only dataset actions are used is also very clever and matches that of conservative offline RL. I also like the idea of $q$ and $v$ values that estimate node and subtree values so that the tree does not need to be fully traversed before an action can be picked.

**Weakness:**\
However, it seems the paper is not complete in several aspects. I believe the paper can benefit from more explanations both theoretically and empirically:
1. the tree essentially acts as a policy to output actions given input states. That means for every input state we should maintain a tree and more importantly have all its node and branch values **accurately estimated**. However, this seems to be prohibitive for continuous states since we need to build infinitely many trees. How did you deal with continuous states? Did you apply some binning? When dealing with continuous spaces, the network function $f(\cdot, \cdot; \theta)$ typically maps to a distribution like Gaussian, and we can generally hope the network generalize to similar states, but I don't see how that is possible given the way the tree was constructed in BraVE.

2. for the tree to be useful, we need to have accurate estimates of $q,v$. But because $v$ is a summary of values of all subtrees, how did BraVE ensure that the values from unchosen subtrees (evetually leaf nodes) were accurate? Did you need extra assumptions or structures on the network $f$ to ensure unchosen leaf nodes had reasonable values? More generally, if $q$ and $v$ had estimation errors, how did the errors influence action selection? Because of the tree structure, I would expect that the chance of picking a wrong action depends on tree level. This type of discussion would add to the contribution of the paper.

3. I am afraid that the experiments are limited in diversity. In the first paragrah of the paper the authors mentioned applications like healthcare but they were not touched in the experiments. Moreover, some related papers tested the discretized continuous action spaces, that would also add to the credibility of the paper, see ref [1].  Related to this, I found it very strange that IQL can perform well even with this large discretized action space in supp? Why this is the case?

References:\
[1] Path Consistency Learning in Tsallis Entropy Regularized MDPs, ICML 2018

---

> ### Author Rebuttal · Authors · 2025-07-30
>
> Thank you for your review. We’re glad you agree that this space is of **"fundamental interest to RL"** and appreciate your kind remarks that our **"tree-based method is principled"** and that our sparsification strategy is **"very clever"** and well-aligned with conservative offline RL. We also appreciate your interest in our $q$ and $\mathbf{v}$ structure, and agree that the ability to select actions without full tree traversal is a key strength.
>
> We are pleased that your concerns primarily relate to clarification, and we have addressed each one with additional explanation and technical detail. We have also conducted a new experiment based on your feedback. We hope these improvements make the paper's contribution and readiness for publication clear. If you find that your concerns have been fully addressed, we would be grateful if you’d consider updating your evaluation. We are happy to continue the discussion if further questions arise. Thank you again for your thoughtful and constructive review.
>
> ### W1: How does BraVE handle continuous state spaces?
>
> BraVE handles continuous state spaces naturally and does not require building infinitely many trees. As described in Section 3.1, the tree is defined over the **discrete action space**, not the state space. The network $f(\theta)$ takes a state $s$ (which may be discrete or continuous) as input and returns Q-values and branch values over the action tree for that specific $s$. Generalization to similar states is achieved through the function approximation capacity of the neural network, as is standard in deep RL. To select an action for any given input state, BraVE performs a single tree traversal guided by the network's outputs for that state. No state binning or discretization is required. We have added the following clarification to Section 3.1:
>
> > At a node $\mathbf{a}\_\text{node}$, the network receives as input the current state $s$, which may be discrete or continuous, and the $N$-dimensional action vector corresponding to $\mathbf{a}\_\text{node}$.
>
> ### W2: How does BraVE ensure accurate estimates for unchosen subtrees?
>
> BraVE does not explicitly ensure the accuracy of unchosen subtrees during a single action-selection pass. Instead, accurate estimates emerge over time as the model is optimized. Over the course of training, BraVE receives supervision across the tree via sampled actions from the offline dataset. The $L_{\text{BraVE}}$ loss (Algorithm 1, Figure 3) propagates value targets upward from sampled actions, so that with sufficient coverage, all relevant nodes and branches receive learning signals. This is analogous to how DQN learns Q-values for actions not selected in a single step, but through repeated sampling over time.
>
> ### W3: Sensitivity to estimation errors at different tree depths
>
> You are correct that estimation errors — particularly those near the root of the tree — can have a disproportionate impact on downstream decision-making. To mitigate this, we proposed two complementary mechanisms which are described in Section 3.5. First, we apply a depth penalty $\delta$, which increases the training signal for nodes closer to the root. This prioritizes the correction of early-stage errors, which influence a larger portion of the tree. Second, at inference time, we use beam search to explore multiple promising paths simultaneously rather than committing to a single greedy trajectory. This improves robustness to local estimation errors and guides more accurate action selection. The impact of these mechanisms is analyzed in our ablation studies (Appendix D); we have also expanded the main text to summarize the key findings.
>
> ### W4: Experimental diversity
>
> Our focus is on introducing and isolating the algorithmic contribution of BraVE — a method designed to address complex sub-action dependencies — in environments that exhibit these structures clearly and controllably. We constructed CoNE specifically for this purpose. In many existing benchmarks (e.g., discretized continuous control), sub-action dependencies are weak or absent [1]. Consequently, they are ill-suited for validating the specific challenge BraVE targets.
>
> ### W5: Why does IQL perform reasonably well despite the large action space?
>
> IQL's relative success, though still significantly outperformed by BraVE (typically by a factor of $2-5\times$), is not entirely surprising. As a strong, general-purpose offline RL algorithm, it makes no structural assumptions and can, in principle, represent any Q-function — unlike factorization-based methods such as FAS. However, this generality comes at a significant cost in modeling complexity, as we detail in Appendix A. BraVE’s tree and sparsification introduce structure and inductive bias that improves efficiency and scalability by guiding learning toward plausible and high-value regions of the action space.
>
> ### W6: Tree traversal relies on a fixed predefined ordering of sub-action dimensions
>
> To assess BraVE’s robustness to the ordering of sub-action dimensions, **we have conducted a new experiment as follows**.
>
> We trained BraVE, FAS, and IQL using five different randomly permuted action orderings in the 8-D, 50% pit CoNE setting, which is empirically the most challenging 8-D variant. The results are shown below:
>
> |        BraVE       |          FAS         |         IQL        |
> | :----------------: | :------------------: | :----------------: |
> | $-53.5 \pm 29.1$ | $-1103.6 \pm 51.4$ | $-96.2 \pm 51.8$ |
>
>
> These findings are consistent with those originally reported in Table 2, showing that BraVE outperforms FAS by an order of magnitude and is roughly twice as effective as IQL. This demonstrates that BraVE’s performance is robust to the ordering of sub-action dimensions and not dependent on a favorable arrangement. We have added this analysis to Section 4.3.
>
> ### Q1: Can this framework support softmax (stochastic) action selection?
>
> Yes — in principle, the tree traversal framework could be adapted to support stochastic policies via softmax selection. Instead of choosing the maximal branch value at each node, one could sample branches according to a softmax over the $\mathbf{v}$ values (e.g., $\text{softmax}(\mathbf{v} / \tau)$ for temperature $\tau$). This results in a stochastic policy over joint actions. In turn, the Bellman target would shift from using $\max_{\mathbf{a}'} Q(s', \mathbf{a}')$ to an expectation under this stochastic policy, effectively turning the update into a soft Q-learning or expected SARSA-style rule.
>
> This would require some modifications to both inference and learning but is entirely compatible with the tree framework. Designing a full actor-critic variant with a parameterized policy is another promising direction for future work. Thank you for the suggestion.
>
> [1] Beeson et al. An investigation of offline reinforcement learning in factorisable action spaces (2024).
>
> [2] Tang et al. Leveraging factored action spaces for efficient offline reinforcement learning in healthcare (2022).

---

> > ### Comment · Reviewer_BzAb · 2025-08-05
> >
> > Thank you for the detailed response.
> >
> > My overall impression towards the paper is quite positive despite there are some unclear points. For example, I understand that the tree is defined on the action space. But acting as a policy $\pi(a|s)$ conditional on state, for each state the tree's nodes needs to be re-evaluated. While it is plausible to claim that the tree implementation follows the standard policy network so that it generalizes to adjacent states similarly, the tree itself does not really reduce computational complexity. Rather, it is achieved by greedy traversal of the tree so there is only one branch expanded each time. I am not very convinced that `BraVE does not explicitly ensure the accuracy of unchosen subtrees during a single action-selection pass. Instead, accurate estimates emerge over time as the model is optimized` can sufficiently address this issue, because the tree structure seems to be more easily influenced. But, above all, I believe these issues do not outweigh the contribution of the paper. And I will consider raising my score if the authors can sufficiently discuss these points in the subsequent version of the paper.

---

> > > ### Author Response · Authors · 2025-08-05
> > >
> > > Thank you for the thoughtful follow-up. We're glad our previous response was helpful and that you have a positive impression of the paper.
> > >
> > > You're right that the computational savings come from the guided traversal rather than the tree structure alone. We have added the following clarification to Section 3.3:
> > >
> > > > This traversal mechanism enables BraVE’s computational efficiency: it evaluates a single node per sub-action dimension, resulting in linear $\mathcal{O}(N)$ complexity — a factor of $\prod_d |A_d|/N$ fewer evaluations than exhaustive scoring. The tree serves as the structural scaffold that makes this targeted, linear-time traversal possible.
> > >
> > > We also agree that further clarification is warranted regarding learning in unvisited subtrees. We’ve added the following to Section 3.4:
> > >
> > > > $L_{\text{BraVE}}$ promotes stable branch value estimates by propagating training signals not just to the node corresponding to the sampled action, but also to every ancestor on its path. Thus, internal nodes — even those not directly sampled — receive gradient updates. Because all branches share a single global Q-network, these updates generalize to similar states and actions, allowing BraVE to form reliable estimates even in parts of the tree that are rarely sampled. This combination of recursive supervision and function approximation provides meaningful learning signals throughout the tree, including regions with limited dataset coverage.
> > >
> > > In addition, the depth penalty $\delta$ is specifically designed to mitigate the sensitivity of the tree structure. We have slightly modified the motivation in Section 3.5 to make this role explicit:
> > >
> > > > To mitigate the sensitivity of the hierarchical structure to inaccurate branch value estimates, we use two complementary mechanisms: a **depth penalty** applied *during training* and **beam search** used *during inference*.
> > >
> > > If these additions resolve your remaining concerns, we hope you’ll consider updating your score. We’re happy to discuss further if any questions remain.

---

> > > > ### Comment · Reviewer_BzAb · 2025-08-06
> > > >
> > > > Thanks for the detailed response. I believe it is crucial that the authors discuss not only merits of the tree construction, but also  downsides such as level-dependent error influence and alike we discussed above. Please make sure to add these clarifying sentences to the paper. I have raised my score.

---

> ### Comment · Area_Chair_Rwo6 · 2025-08-04
> **Please engage with the author response**
>
> Please take advantage of the brief time that remains to ask the authors for any additional clarifications.

---

### Official Review · Reviewer_2ATV · 2025-06-22

**Clarity:** 4
**Significance:** 3
**Originality:** 4
**Rating:** 5
**Confidence:** 4

**Summary:**

In this paper, the authors address the problem of offline reinforcement learning in environment with combinatorial action spaces. The proposed method represent the action space with a tree structure where each node represents a complete action and each level correspond to an action dimension.  Each node is characterized by a value estimate and a vector of branch values for each subtree, these values are estimated by a neural network. The neural network is trained using a regularized TD learning loss as well as a custom branch value supervision loss.

The resulting algorithm is evaluated on a synthetic dataset from a n-dimensional navigation problem against an offline RL baseline (IQL) and an offline RL with value factorization (FAS).

The proposed method has similar performance as FAS for low dimensional action space but outperforms both baselines by a lot as the action space grows.

**Questions:**

-	Why a tree and not a graph?
-	What is the definition of $\hat{a}$?
-	Could you provide more intuition about why the reward cannot be factorized in your experiments?
-	What is the impact of the order of the action dimension when constructing the tree in BRAVE?
-	How would BRAVE perform on a non synthetic dataset?

**Ethical Concerns:**

["NO or VERY MINOR ethics concerns only"]

**Final Justification:**

I raised my score to accept after reading the author's rebuttal. I believe the tree based approach is original and interesting, and the problems of combinatorial offline RL that could be addressed are fairly wide. The authors provided reasonable justifications for the weaknesses that I mentioned. One could criticize that the experiments are a bit limited in scope (synthetic environments), but I would argue that it is also a strength of this paper, they designed a well controlled environment to truly understand the performance gap between the proposed method and the baseline.

**Limitations:**

Most of the limitations are acknowledged by the authors:

-	Discrete actions required
-	The ordering of the action can impact the performance (although no guidance is provided on that)
-	It is an offline method, so it depends on the quality of the dataset

**Paper Formatting Concerns:**

no formatting concerns

**Quality:**

4

**Strengths And Weaknesses:**

**Strengths**

- The idea of representing the value function by a tree is interesting and as far as I know novel and present a strong contirbution.

- The paper is well organized, and fairly easy to follow (except the section on the loss calculation).

- The method shows strong performance on the synthetic dataset compared to the baseline. The experiments seem well-designed and they evaluate the method in controlled settings. The results are quite convincing. Some of the description of how the environments are generated could be improved.

**Weaknesses**

1. Why only evaluating on synthetic data? I understand the importance of the experiments proposed to evaluate the algorithm in a controlled setting but it would have made an even stronger case to add an example on a real dataset.

2. The ablation study is an important part of the analysis and lack discussions. The authors provide curves in appendix but do not discuss the results. I believe such a discussion should be included to the main text.

3. Another aspect that is missing and I think crucial is the analysis of the order of the tree. It seems to me that the way the tree is constructed can impact the result greatly, as some actions might be more important than others, if they appear very deep in the tree the performance might be impacted. In the related work discussion MCTS it is mentioned that a sequential structure need to be imposed, however it seems to me that the tree constructed by brave also need to impose some arbitrary structure on the action space. The authors acknowledge this limitation in the conclusion but still I think it could use a dedicated experiment.

4. Explanation of the algorithm sometimes lack clarity

- The loss computation is a key component of the method but lack some clarity. $\hat{a}$? is not defined, and it is not described how $v$ is learned.
- I also did not understand why v, the best value of each subtree, needs to be learned and not computed after the training from the estimated q values. The pseudo code provides a little bit more insights but it would help to make the text consistent.
- Some of the explanation of the results could use more justification, for example when the authors explain the behavior of FAS on high dimensional problem l258.
- The experimental setup with sub-action dependencies is hard to understand. The number of pits is said to increase to 100%, does that mean the whole state space is a pit?

---

> ### Author Rebuttal · Authors · 2025-07-30
>
> Thank you for your thoughtful review. We're excited you found our work to be a **"novel"** and **"strong contribution"**. We also appreciate your comments that our experiments are **"well-designed"** and that our **"results are quite convincing"**.
>
> We’re glad to see that many of your remaining concerns focus on clarification and additional detail — particularly around the loss, ablations, and tree ordering — and we are pleased to report that these have been fully addressed through improved explanations and new experiments. If you feel these updates resolve your concerns, we hope you will consider updating your evaluation accordingly. We would be glad to continue the discussion should any further questions arise. Thank you again for your thoughtful review.
>
> ### W1: Evaluation limited to synthetic data
>
> We believe synthetic environments are sufficient for this work because our goal is to isolate and analyze sub-action dependencies under controlled conditions. While real-world offline RL evaluations may be ideal, they typically rely on off-policy evaluation (OPE), which is known to be noisy and biased. Prior work such as Tang et al. [1] used OPE and expert evaluation for domain-specific insights, which is appropriate for their application-based paper. However, for an algorithmic contribution like ours, we find that a high-fidelity simulator enables direct measurement of policy performance via ground-truth rollouts, providing a clearer and more reliable evaluation signal and ultimately a deeper understanding of BraVE’s strengths and weaknesses compared to SOTA methods.
>
> We view real-world deployment and evaluation as important future work and have added this text to the conclusion (Section 6):
>
> > While our current evaluation uses controlled synthetic environments to enable clear algorithmic analysis, real-world deployment and evaluation is an important direction for future work.
>
> ### W2: Ablation discussion should be in the main text
>
> Thank you for this suggestion. We agree, and have revised Section 4.2 to summarize the key findings from Appendix B as follows:
>
> > We conduct a series of ablation and hyperparameter studies to assess the contribution of BraVE's design decisions. First, we examine the effect of the depth penalty $\delta$, which scales the loss based on a node's position in the tree. A small penalty ($\delta=1$) consistently yields the best results, confirming the importance of emphasizing accuracy near the root. Second, we vary the loss weight $\alpha$ to examine the trade-off between the behavior-regularized TD loss and the BraVE loss. While performance is stable across a range of $\alpha$ values, omitting the TD term entirely ($\alpha=0$) leads to degradation, particularly in higher-dimensional tasks. Third, to isolate the role of BraVE's tree structure, we compare against a DQN baseline trained with the same loss but without hierarchical traversal or branch value propagation. This variant fails to learn viable policies, highlighting the benefits of BraVE's structured decomposition. Fourth, we evaluate the effect of beam width on performance and inference time. A width of 10 is sufficient for strong performance, and even large widths incur minimal runtime overhead, demonstrating the method's practical efficiency. Finally, we assess BraVE's sensitivity to sub-action dimension ordering. We retrain BraVE, FAS, and IQL across five random action orderings in the most challenging 8-D, 50% pit setting. Results remain consistent with our main findings (Table 2), confirming that BraVE's performance does not depend on a favorable ordering. Full results are presented in Appendix D.
>
> We thank you for encouraging this improvement. Note that the action order ablation was added in direct response to your feedback and is further discussed in W3.
>
> ### W3: Missing analysis of action dimension ordering
>
> We fully agree that this analysis is important and **we have conducted a new experiment to evaluate BraVE's sensitivity to action dimension ordering as follows**.
>
> We trained BraVE, FAS, and IQL using five different randomly permuted action orderings in the 8-D, 50% pit CoNE setting, which is empirically the most challenging 8-D variant. The results are shown below:
>
> |        BraVE       |          FAS         |         IQL        |
> | :----------------: | :------------------: | :----------------: |
> | $-53.5 \pm 29.1$ | $-1103.6 \pm 51.4$ | $-96.2 \pm 51.8$ |
>
>
> These findings are consistent with those originally reported in Table 2, showing that BraVE outperforms FAS by an order of magnitude and is roughly twice as effective as IQL. This demonstrates that BraVE’s performance is robust to the ordering of sub-action dimensions and not dependent on a favorable arrangement. We have added this analysis to Section 4.3.
>
> ### W4: $\hat{\mathbf{a}}'$ is undefined
>
> The variable $\hat{\mathbf{a}}'$ is defined on lines 138 and 171, as well as in the Require section of Algorithm 1, as the action selected via tree traversal. Please let us know if you believe it would be helpful to reiterate this definition elsewhere in the text.
>
> ### W5: It is unclear why and how $\mathbf{v}$ is learned
>
> The greedy action $\hat{\mathbf{a}}'$ is needed to compute the Bellman target for updating $Q(s, \mathbf{a})$ (Equation 1). During the forward pass, however, the Q-values for all descendant nodes are not available, as they are themselves outputs of the network being trained. To avoid evaluating the full combinatorial action space at each step, we rely on the network's estimates of $\mathbf{v}$ to guide tree traversal toward promising branches. In this way, $\mathbf{v}$ is fundamental to learning $Q$, and thus cannot be computed post hoc from already-trained Q-values. We have clarified in Section 3.3 as:
>
> > Crucially, these branch values must be learned and predicted by the network, as they guide the efficient search for$\hat{\mathbf{a}}'$ used in the Bellman target computation (Equation 1), before the values of descendant nodes are themselves known.
>
> As described beginning on Line 179 and illustrated in Figure 3, $\mathbf{v}$ is learned via the $L_{\text{BraVE}}$ loss, which recursively enforces consistency between branch values and the maximum Q-value of a node's descendants. We have clarified this training mechanism in Section 3.4:
>
> > In this way, the propagated targets directly supervise the network's predictions for the branch values $\mathbf{v}$ at each node. Thus, $\mathbf{v}$ is explicitly learned through $L_{\text{BraVE}}$ to reflect the highest Q-value available in each subtree.
>
> ### W6: It is unclear why FAS's performance degrades in high-dimensional problems when rewards are non-factorizable
>
> FAS’s performance degrades in this setting due to its fundamental simplifying assumption of linear value decomposition in the reward $r(s, a) = \sum_{d=1}^D r_d(s, a_d)$.
>
> More specifically, the reward function $r = -\text{distance}(s, g)$ depends on the joint effect of all sub-actions and cannot be decomposed additively. For example, the joint action “move up and left” leads to a diagonal movement with a specific reward. Evaluating “move up” and “move left” separately would yield different states and thus different rewards, breaking the additive assumption (see Appendix A). We have clarified this reasoning in Section 4.1 to better support the empirical findings:
>
> > Because the reward is computed from the final state resulting from the full action vector, it is not meaningful to assign individual rewards to sub-actions in isolation. This mismatch introduces a modeling bias into FAS that grows with action dimensionality. As a result, FAS learns inaccurate Q-values and exhibits degraded performance. BraVE avoids this instability by evaluating full joint actions directly, maintaining high performance across dimensions.
>
> ### W7: It is unclear if 100% pit coverage in Table 2 means the whole state space is a pit
>
> As noted on line 266, 100% pit density refers to the fraction of *interior* (non-boundary) states that are designated as pits. To make this more explicit, we have clarified the text in Section 4.1 as follows:
>
> > In an 8-D environment (\$|\mathcal{A}| = 65,536\$), we vary the number of pits from 5% to 100% of the 6,561 interior (non-boundary) states, ensuring that there is always a feasible path to the goal along the boundary of the state space. Boundary states are never assigned as pits.
>
> ### Q1: Why a tree and not a graph?
>
> The tree provides a structured framework for optimizing over high-dimensional combinatorial actions. It partitions the joint action space by progressively fixing sub-action components at each level, enabling efficient traversal and targeted evaluation of full action candidates. This structure supports recursive training of both Q-values and subtree maxima via the $L_{\text{BraVE}}$ loss.
>
> While a graph could offer similar structural benefits, it introduces cycles and multiple paths between nodes, which complicates inference and learning by making value propagation more difficult. The tree, by contrast, offers a clean and tractable structure that aligns well with our optimization objective. Our results confirm the practical effectiveness of this approach.
>
> ### Q2: What is the definition of $\hat{a}'$?
>
> Please see W4.
>
> ### Q3: Why is the reward non-factorizable?
>
> Please see W6. The reward $r = -\text{distance}(s, g)$ is non-factorizable because it depends on the joint outcome of the full action vector and cannot be expressed as a sum of rewards for individual sub-actions.
>
> ### Q4: What is the impact of action ordering?
>
> Please see W3. Our new experiments demonstrate that BraVE is robust to the ordering of sub-actions.
>
> ### Q5: How would BraVE perform on a non-synthetic dataset?
>
> Please see W1.
>
> [1] Tang et al. Leveraging factored action spaces for efficient offline reinforcement learning in healthcare (2022).

---

> > ### Comment · Reviewer_2ATV · 2025-08-04
> >
> > Thank you for the thorough response to my comments. I appreciate that you ran an extra experiment to study the impact of action ordering and it is great to know that in this environment the method seems robust to the ordering.
> >
> > Apologies that I missed the definition of $\hat{a}'$.
> >
> > The motivation for learning $v$ is also clearer now.
> >
> > Regarding the discussion on real world dataset, I understand your concern on OPE being noisy. Would there be any extra difficulty of using the tree structured value function for OPE?

---

> > > ### Author Response · Authors · 2025-08-04
> > >
> > > Thanks for your continued engagement — we're glad to have clarified your concerns.
> > >
> > > That's a great question about OPE. Since it could refer either to evaluating BraVE-learned policies or to extending OPE methods using BraVE's tree structure, we address both perspectives below.
> > >
> > > From the perspective of *evaluating* policies learned via BraVE, **no, BraVE's tree structure does not introduce any additional difficulty beyond that of other offline RL methods.** After learning, BraVE exposes a conventional Q-function, $Q(s, a)$, so any standard OPE method can be applied in principle.
> > >
> > > From the perspective of *extending* existing OPE methods, BraVE's tree structure presents a promising direction for future work. Because the tree contains only actions observed in the dataset, the evaluation policy never selects a joint action with zero probability under the behavior policy. This avoids the infinite or numerically unstable importance weights that often dominate variance in high-dimensional spaces, and suggests a straightforward path toward more stable, structure-aware OPE. Although reducing variance in OPE via tree-structured action spaces is non-trivial and a fundamentally distinct line of research, it is a compelling direction that we are actively pursuing as part of ongoing work.
> > >
> > > If these updates address your concerns, we hope you'll consider updating your evaluation. We're happy to answer any additional questions should they come up.

---

> > > > ### Comment · Reviewer_2ATV · 2025-08-05
> > > >
> > > > That makes sense, I look forward to any future application of the methods on real datasets then.
> > > > Interesting insights on the variance reduction.

---

### Official Review · Reviewer_cJdX · 2025-06-27

**Clarity:** 3
**Significance:** 3
**Originality:** 3
**Rating:** 4
**Confidence:** 4

**Summary:**

This paper introduces Brave, a novel offline RL algorithm designed to handle high-dimensional, discrete combinatorial action spaces. The paper says that existing methods are either computationally infeasible due to exhaustive evaluation of vast number of actions, or sacrifices accuracy by factorizing Q-values. Brave addresses this trade-off by imposing a tree structure on the action space, allowing it to evaluate maxQ with a linear number of evaluations while preserving the dependency structures of action dimensions. The method employs a Q-guidance to traverse this tree, and these guidance are trained with a behavior-regularized temporal difference loss. The experiments show Brave is significantly outperforming baselines.

**Questions:**

- Can we simply use BraVE as an online algorithm if we do not put l2-behavior-regularization? If yes, why should the paper focus on the offline setting?
- By having additional binary indicator of whether the action is selected or not, Q-function of BraVE seems to have different neural architecture compared to other algorithms. How is this implemented, and how do you ensure a fair comparison with other baselines?
- As difficulty of the problem (|A| or |S|) increases, the experiments result does not have a clear trend. For example in table 3, FAS shows much better performance on |S|=15625 compared to |S|=25. What is the reason to this?

**Ethical Concerns:**

["NO or VERY MINOR ethics concerns only"]

**Final Justification:**

I understand the point of the authors, and some of my reviews might be mis-claiming to certain extent. However, I do not think the paper is as "strong" as score 5, considering its weak presentation and experimentations.

**Limitations:**

yes

**Quality:**

2

**Strengths And Weaknesses:**

Strengths
- Proposes a novel method to cleverly implement Q-learning on large combinatorial action space
- Algorithm well explained
- Significantly outperforms other baselines

Weaknesses
- While the proposed method is about handling large combinatorial action space, the paper limits its scope on offline RL settings, and the contribution specific to offline setting is minor (adding l2 regularization). Considering that the main contribution can be applied to any RL settings, I believe that the paper should have been focused on standard online setting rather than offline setting.
- While the paper only says about the FAS approach in the paper, actually there have been a number of different approaches toward large combinatorial action space. If we insist learning with Q-learning-based algorithms, we can either learn an action embedding [1], or use a model-based approach through Q=r+V [2].
[1] Dulac-Arnold, Gabriel, et al. "Deep reinforcement learning in large discrete action spaces." (2015)
[2] Vinyals, Oriol, et al. "Grandmaster level in StarCraft II using multi-agent reinforcement learning." nature 575.7782 (2019): 350-354.
  - If we are OK with other type of algorithms, policy-based algorithms (e.g., PPO) or sequence modelling approaches (e.g., Decision Transformer) will be able to easily solve the problem.
  - Therefore, I am not very satisfied with the empirical evaluations done in the paper; it is even very hard to get a feeling about whether Brave is working or not, as FAS and IQL are mostly broken and any other types of algorithm, or oracle policy, are not included.
- Furthermore, all three algorithms experimented have a very different behavior regularizations. IQL - quantile regression, FAS - BCQ like, BraVE - l2 regularization. While the difference between BraVE and FAS on how to handle large combinatorial action space is the main focus here, some of the performance differences may have been originated from these different behavior regularizations; I would say this is not a good way to demonstrate the performance of the proposed method.

---

> ### Author Rebuttal · Authors · 2025-07-30
>
> Thank you for your review.  We are overall pleased that you recognize BraVE as a **"novel"** method that **"cleverly implements Q-learning on large combinatorial action space"** and how it **"significantly outperform[s] baselines."**
>
> Several of your concerns appear to reflect an online RL perspective. However, our method is explicitly designed for the offline setting, with contributions motivated by challenges specific to that regime. These points can be resolved by increasing the clarity of the paper, as outlined point-by-point below. We hope these clarifications warrant a reconsideration of your score. Thank you again for your thoughtful and constructive review.
>
> ### W1: The paper limits its scope to offline RL, while the method could be applied online
>
> We appreciate the opportunity to clarify our work's positioning. Our focus on offline RL is deliberate and central to our contribution, which goes well beyond simple L2 regularization.
>
> First, BraVE introduces a tree sparsification mechanism (Section 3.2) that is only possible in the offline setting. It uses the fixed dataset $\mathcal{B}$ to prune the action tree, restricting Q-value estimation to actions with empirical support. This serves as a structural regularizer that mitigates overestimation from out-of-distribution actions — a key challenge in offline RL that does not arise in the online setting.
>
> Second, we focus on the offline setting because of its practical importance. As stated in our introduction, many domains with combinatorial action spaces, such as treatment planning, resource allocation, or industrial control, preclude online exploration due to safety, ethical, or cost constraints.
>
> Offline RL has emerged as a distinct field in which methods are designed precisely to address these challenges [1]. Modern offline algorithms, such as BCQ [2] and IQL [3], are defined by their approaches to the distribution shift caused by fixed data. Our work is situated within this paradigm, offering a new structural approach to a well-established offline RL problem.
>
> While core ideas from BraVE may be adaptable to the online setting, our work is specifically designed for offline RL — a distinct setting with its own challenges.
>
> To resolve your concern, we have updated the Discussion and Conclusion (Section 6) by adding the following:
>
> > Finally, while our work focuses on the offline setting, we believe BraVE provides a strong foundation for online combinatorial RL. Extending our framework would require addressing a new form of exploration–exploitation trade-off, potentially by replacing our dataset-driven tree sparsification with a dynamic mechanism for growing and pruning the action tree during learning.
>
> ### W2: Comparison to other approaches for large combinatorial action spaces
>
> Thank you for these suggestions. We compare BraVE against the **most relevant and state-of-the-art baselines under the offline RL setting**. Our choice of baselines is principled and more comprehensive than related works’, as detailed below.
>
> First, we note that Dulac-Arnold et al. [4] is already cited in Section 5.2 as a foundational method for large discrete action spaces. However, it is inherently an **online** algorithm with no straightforward extension to offline RL. Similarly, PPO is an on-policy online method, and thus not directly applicable to our setting. While Decision Transformer is a valid offline approach, it reframes RL as sequence modeling — conceptually different from value-based learning. Since our work is situated in the value-based paradigm, we compare against IQL (SOTA in value-based offline RL) and FAS (SOTA in combinatorial spaces). Demonstrating that BraVE overcomes the failure modes of these widely-used methods is, we believe, a more direct and informative contribution.
>
> Our evaluation is also more thorough than many recent works in this space. For instance, Tang et al. [5] (FAS) compared only against BCQ, while Beeson et al. [6] compared multiple factored methods against each other and an oracle, but not against non-factored approaches like IQL. By comparing BraVE against both a strong factored baseline (FAS) and a non-factored baseline (IQL), we provide a more complete and rigorous characterization of BraVE's advantages.
>
> ### W3: It is difficult to assess BraVE's performance because FAS and IQL are "mostly broken" and an oracle policy is not included
>
> We respectfully but firmly disagree.
>
> BraVE was explicitly designed to address the failure modes of existing offline RL methods in the presence of sub-action dependencies. The CoNE benchmark was constructed to isolate and control this challenge. Our hypothesis — that standard algorithms like IQL and FAS would fail under these conditions — is strongly validated by the observed empirical results.
>
> That these baselines perform poorly is not a shortcoming of our evaluation; it is a **core experimental finding** and a primary motivation for BraVE. Highlighting these failure cases is critical for progress in this area, and underscores the need for approaches that go beyond naive factorization or flat value estimation.
>
> To further contextualize performance and address the concern that an oracle policy is not included, **we compute a new metric, the normalized episode-length score, to measure each method's performance relative to an oracle baseline for the results in Table 2**. This score places each method on a 0–100 scale, where 0 corresponds to a random policy and 100 to an oracle planner. It is defined as:
>
> $$\text{score}\_\text{norm} = 100 \cdot \frac{\text{length}\_\text{random} - \text{length}\_\text{agent}}{\text{length}\_\text{random} - \text{length}\_\text{oracle}}$$
>
> where $\text{length}\_\text{agent}$ is the average episode length of the evaluated policy, $\text{length}\_\text{random}$ is the episode length of a random policy (which consistently times out at 100 steps), and $\text{length}\_\text{oracle}$ is the optimal path length computed via A$^*$ search.
>
> Below are the scores for the setting with non-factorizable rewards and sub-action dependencies (Table 2):
>
> | Pit % | BraVE |  FAS |  IQL |
> | --- | :---: | :--: | :--: |
> | 0     |  99.3 | 89.6 | 88.0 |
> | 5     |  92.8 | 57.3 | 52.9 |
> | 10    |  94.1 | 20.2 | 51.5 |
> | 25    |  78.5 |  0.0 | 54.7 |
> | 50    |  71.4 |  0.0 | 40.6 |
> | 75    |  78.5 |  0.0 | 38.4 |
> | 100   |  79.3 |  0.0 | 37.0 |
>
> These results show that BraVE maintains strong performance as sub-action dependencies become more severe, precisely the regime where prior methods fail. This validates both CoNE as a benchmark and BraVE as an effective solution.
>
> ### W4: Performance differences may stem from differences in behavior regularization
>
> We appreciate this concern. Our comparison used the published, SOTA versions of IQL and FAS, which include their respective behavior regularization mechanisms: expectile regression in IQL, and BCQ-style filtering in FAS. These are not auxiliary add-ons, but integral components of each algorithm. Creating custom versions of these methods using a shared regularizer (e.g., our L2 penalty) would introduce untested variants, undermining reproducibility and fairness. We believe that comparing BraVE against strong baselines as defined in prior work is the most standard and principled approach.
>
> Moreover, the large performance gaps (e.g., $-41.6$ vs. $-1131.4$), are unlikely to be caused by regularization alone and instead reflect BraVE's superior modeling of sub-action dependencies.
>
> ### Q1: Can BraVE be used as an online algorithm if we omit the L2 regularization?
>
> Please see W1.
>
> ### Q2: Does BraVE use a different neural architecture (e.g., a binary indicator)?
>
> BraVE does not use binary indicators or encode any tree-specific structure into the Q-network input. The Q-function is implemented as a standard MLP that takes the full state $s$ and full action vector $a$ as input — the same input format used by IQL.
>
> The tree is used solely to structure the $\arg\max$ operation during inference and to define the recursive loss used to train the $\mathbf{v}$ values. It does not modify the architecture or provide additional context to the Q-function itself.
>
> We have clarified this in Section 3.3 to make the model architecture and input specification more explicit:
>
> > Importantly, the network $f$ is a standard function approximator (e.g., an MLP) and does not require any specialized architecture. It takes as input the state $s$ and the complete action vector $\mathbf{a}_\text{node}$. The tree is not encoded into the network input; instead, it is an external structure that organizes the $\arg\max$ computation and defines the recursive supervision used to train the branch values $\mathbf{v}$.
>
> ### Q3: Why does FAS perform better on $|\mathcal{S}| = 15625$ than on $|\mathcal{S}| = 25$?
>
> In Table 3, the number of pits is fixed at 5. As the state space grows, these pits comprise a smaller fraction of the environment. FAS's apparent improvement reflects this reduced exposure, not an improved ability to handle hazardous interactions.
>
> BraVE, by contrast, maintains consistent performance across environment sizes because it models sub-action dependencies correctly, regardless of how frequently they arise. This contrast highlights BraVE's robustness, even when rare events play a critical role in task success.
>
> [1] Levine et al. Offline reinforcement learning: Tutorial, review, and perspectives on open problems (2020).
>
> [2] Fujimoto et al. Off-policy deep reinforcement learning without exploration (2019).
>
> [3] Kostrikov et al. Offline reinforcement learning with implicit q-learning (2021).
>
> [4] Dulac-Arnold et al. Deep reinforcement learning in large discrete action spaces (2015).
>
> [5] Tang et al. Leveraging factored action spaces for efficient offline reinforcement learning in healthcare (2022).
>
> [6] Beeson et al. An investigation of offline reinforcement learning in factorisable action spaces (2024).

---

> > ### Comment · Reviewer_cJdX · 2025-08-03
> >
> > Thank you for addressing my concerns. I raised my score.

---

### Official Review · Reviewer_KDjC · 2025-07-02

**Clarity:** 4
**Significance:** 3
**Originality:** 4
**Rating:** 5
**Confidence:** 4

**Summary:**

Authors introduce BraVE an offline RL algorithm for combinatorial discrete action spaces with multiple dimensions related to different sub-actions.

The problem of evaluating the action-value function maximum over a large combinatorial action space $\max_a Q(s,a)$ (given a state $s$) needed for action-value function learning using TD method is alleviated by using dedicated neural network $f$ and organizing
actions into a tree structure where, given some fixed ordering over sub-action dimensions, in each tree level one sub-action is decided.

The trick is that the network $f(s,a)$ is trained to output $Q(s,a)$, where $a$ corresponds to some tree node, but also to output maximum values over sub-trees corresponding to all its child nodes.

This is done by updating $f$ loss with every $Q$ target computation.

Such $f$ then allows for fast $\max_a Q(s,a)$ evaluation (linear in number of action space dimensions).

Besides this basic idea authors include many practical improvements  including:
tree sparsification (to include only nodes which corresponds to actions actually present in the off-line dataset),
masking (when number of sub-actions across sub-action dimensions varies),
depth-penalty weight (helping to weight more the errors closer to the root node where decisions are more critical),
and beam search (for better robustness during inference).

Authors demonstrate feasibility of the method on their proposed benchmark which allows for strongly dependent sub-actions (depending on a given state) with superior performance over compared baselines.

**Questions:**

1) I am missing information on how the $f$ network is initialized
in order to fit with $\max_a Q(s,a)$ evaluation. One could  train it with some lower bound on $Q$ values, but it seems to introduce an inconvenient bias. Could you clarify this?

2) From Figure 13a, it seems that increasing beam width brings rather problems (decrease in return). Could you comment on this?

**Ethical Concerns:**

["NO or VERY MINOR ethics concerns only"]

**Final Justification:**

The paper approach is novel addressing a gap in current RL portfolio (combinatorial discrete action spaces with dependent sub-action in offline setting). The authors clarified all my questions. I also read the other reviews. As a result I think the paper should be accepted.

**Limitations:**

Yes

**Paper Formatting Concerns:**

I have found no major formatting issues.

**Quality:**

3

**Strengths And Weaknesses:**

The paper is well written, the presentation is clear, and the proposed method is well motivated.

Although there is not much math, I found no errors and typos in the math formulas.

I found the proposed idea of how to cope with $\max_a Q(s,a)$ evaluation compelling.

I would  like to see the generalization of the proposed method to the standard online RL setting where probably also tree structure has to be updated due to possible distribution shift.

Generally, I have no complaints about the paper.

---

> ### Author Rebuttal · Authors · 2025-07-30
>
> Thank you for your very positive review. We are delighted that you found our core idea **"compelling"** and **"generally have no complaints about the paper"**. We also appreciate your recognition of the **"practical improvements"** that strengthen BraVE beyond its core tree-based traversal framework. We are grateful for your confidence in our work. Below, we address your questions and comments in more detail.
>
> ### Q1: How is network initialized in order to fit with $\max_a Q(s,a)$ evaluation?
>
> We use standard initialization (Kaiming) as in a typical DQN. No special treatment is needed: both node Q-values ($q$) and branch values ($\mathbf{v}$) are learned jointly via Bellman supervision and the $L_\text{BraVE}$ loss (Algorithm 1, Figure 3). The branch values are not pre-initialized or pre-trained. Instead, $L_\text{BraVE}$ drives them to match the maximum of their children, enabling structure to emerge from data without relying on priors or artificial constraints.
> ### Q2: Why does increasing beam width sometimes decrease return?
> This is an excellent observation. Beyond a beam width of ~50, performance can degrade because broader searches enter parts of the tree where Q-values are less reliable. Narrower beams implicitly regularize by keeping search within high-confidence regions.
> To ensure this point is clear in the paper, we have added the following clarification to Section 4.3:
> > This degradation arises because broader beams may explore low-confidence regions of the tree where Q-values are less reliable. Narrower beams serve as a form of regularization, restricting search to well-learned regions of the action space.
> Since beam search is used only at inference time, beam width remains a tunable hyperparameter that can be adjusted post hoc based on compute or task requirements.
> ### Q3: How might BraVE extend to the standard online RL setting?
>
> We agree that the online setting is an exciting and important direction for future work. Enabling online use would indeed require an alternative mechanism (such as dynamic tree construction), since our current approach builds and sparsifies the tree using a fixed offline dataset $\mathcal{B}$ (Sections 3.1-3.2).
>
> To help point others in this direction, we have added the following sentence to the Discussion and Conclusion (Section 6) describing BraVE’s potential relevance beyond the offline RL:
>
> > Finally, while our work focuses on the offline setting, we believe BraVE provides a strong foundation for online combinatorial RL. Extending our framework would require addressing a new form of exploration–exploitation trade-off, potentially by replacing our dataset-driven tree sparsification with a dynamic mechanism for growing and pruning the action tree during learning.

---

> > ### Comment · Reviewer_KDjC · 2025-08-04
> >
> > I would like to thank the auhors for addressing my questions. I have read also the other reviews. As a result I decided to keep my score/rating.

---

> ### Comment · Area_Chair_Rwo6 · 2025-08-04
> **Do you have any additional questions?**
>
> The authors provided a brief response. Do you have any additional questions for them?

---

### Note · Authors · 2025-08-12

We thank the reviewers for the productive review process. We are pleased to have received such a positive reception, with reviewers  finding BraVE to be a **"strong contribution"** and a **"novel," "compelling,"** and **"principled"** method that **"significantly outperforms baselines"** while addressing a problem of **"fundamental interest to RL."**

Rebuttal discussions focused on clarifying experimental scope, robustness, and algorithmic details. We addressed these points in four ways:

(1) **New Experiment**: Adding new requested experiment confirming robustness to action ordering and summarizing key ablations in the main text;

(2) **Strengthened Evaluation**: Introducing oracle-normalized scores to provide a more rigorous and grounded comparison of performance against baselines;

(3) **Clarifications**: Clarifying the loss computation, the role and learning of branch values, handling of continuous states, and computational efficiency; and

(4) **Expanded Discussion**: Motivating BraVE’s offline focus, describing the relationship to the online setting, and potential extensions.

These updates resolved the reviewers’ core issues. Consequently, **three of the four reviewers increased their scores/ratings after the rebuttal (the remaining reviewer had already given a "5 – Accept").** The final reviews reflect a clear consensus that BraVE is a technically solid, empirically strong, and original contribution to offline RL.

---

### Decision · Program_Chairs · 2025-09-17

**Decision:**

Accept (poster)

**Comment:**

After some extensive discussion with the authors, the reviewers converged on a positive assessment of the paper. There were no concerns raised about the ability of the authors to modify their paper to incorporate lessons learned from the discussion, so my meta-review is positive.

One suggestion for the authors: While your focus is on combinatorial actions, it still might be appropriate to review some of the work on continuous actions since the techniques are not so different and could be applied to combinatorial domains with little or no modification. See, for example, the work of Jason Pazis, which explored both tree in a Q-learning setting, and other methods of splitting the action space in the case of approximate linear programming. (Please note, this is not a complaint or accusation that your work is not original, but genuinely meant as something you might find interesting if you haven’t already explored it.)